# Geometric Autoencoder Priors for Bayesian Inversion: Learn First Observe Later

**Arnaud Vadeboncoeur**[*]
Department of Engineering
University of Cambridge
Cambridge, 7a JJ Thomson Ave, UK
`av537@cam.ac.uk`

**Gregory Duthé**[*]
Institute of Structural Engineering
ETH Zürich
Zürich, Stefano-Franscini-Platz 5, Switzerland
`duthe@ibk.baug.ethz.ch`

**Mark Girolami**
Department of Engineering
University of Cambridge
Cambridge, 7a JJ Thomson Ave, UK
`mag92@cam.ac.uk`

**Eleni Chatzi**
Institute of Structural Engineering
ETH Zürich
Zürich, Stefano-Franscini-Platz 5, Switzerland
`chatzi@ibk.baug.ethz.ch`

## Abstract

Uncertainty Quantification (UQ) is paramount for inference in engineering. A common inference task is to recover full-field information of physical systems from a small number of noisy observations, a usually highly ill-posed problem. Sharing information from multiple distinct yet related physical systems can alleviate this ill-posedness. Critically, engineering systems often have complicated variable geometries prohibiting the use of standard multi-system Bayesian UQ. In this work, we introduce Geometric Autoencoders for Bayesian Inversion (GABI), a framework for learning geometry-aware generative models of physical responses that serve as highly informative geometry-conditioned priors for Bayesian inversion. Following a "learn first, observe later" paradigm, GABI distills information from large datasets of systems with varying geometries, without requiring knowledge of governing PDEs, boundary conditions, or observation processes, into a rich latent prior. At inference time, this prior is seamlessly combined with the likelihood of a specific observation process, yielding a geometry-adapted posterior distribution. Our proposed framework is architecture-agnostic. A creative use of Approximate Bayesian Computation (ABC) sampling yields an efficient implementation that utilizes modern GPU hardware. We test our method on: steady-state heat over rectangular domains; Reynolds-Averaged Navier-Stokes (RANS) flow around airfoils; Helmholtz resonance and source localization on 3D car bodies; RANS airflow over terrain. We find: the predictive accuracy to be comparable to deterministic supervised learning approaches in the restricted setting where supervised learning is applicable; UQ to be well calibrated and robust on challenging problems with complex geometries.

## 1 Introduction

Many important problems in engineering deal with spatially varying quantities that must be inferred. The task of inferring full-field information from sparse noisy measurement is a type of inverse problem (Kaipio & Somersalo, 2005); these often need to be regularized due to their inherent ill-posedness (Calvetti & Reichel, 2003). Statistically interpretable regularization can be achieved through the Bayesian paradigm (Stuart, 2010), where we combine a likelihood function characterizing data fit for a particular observation process, and a prior which characterizes assumptions about the data-generating process. Bayesian inference allows for robust and principled uncertainty quantification, where we obtain a distribution on the estimated quantity of interest given data from a particular physical system. The main challenge in enacting a Bayesian program is prior specification; often, not

---

[*]Corresponding Authors; Code: `https://github.com/gduthe/GABI`

much can be said about the prior which often leads to diffuse and vague posteriors with relatively poor predictive accuracy when compared to deterministic methods. To alleviate this vagueness, one possible strategy is to learn a suitable prior from a dataset of related but distinct physical setups, where more data may be available. This is explored in general-purpose AI applications (Shwartz-Ziv et al., 2022; Feng et al., 2023; Boys et al., 2024) and physical applications (Meng et al., 2022; Akyildiz et al., 2025; Patel & Oberai, 2021; Patel et al., 2022; Laloy et al., 2018; 2017). However, in engineering, *geometry* plays a crucial role in determining the dynamics arising for a physical system. Hence, for truly general purpose, practical methodology, learned priors must incorporate knowledge about the geometry of the problem at hand. But changing geometries hinder the direct learning of a prior over fields as the probability spaces involved are tied to the geometry of each individual physical system, prohibiting a direct treatment.

In this work, we propose a methodology using graph-based autoencoders to learn highly informative priors as geometry-aware generative models for use in Bayesian inversion tasks. We also show how to effectively solve the Bayesian inverse problems resulting from these priors through a pushforward model. Importantly, the proposed scheme decouples the task of "learning" and the task of "inferring from observations", resulting in a train-once use-everywhere approach which is independent of the particular observation process for solving inverse problems, a trait not shared by related supervised learning approaches. We make the following contributions:

(C1) We propose a methodology to encode geometries and physical fields into a learned latent prior without specifying the governing PDE or boundary conditions.

(C2) We prove that solving the Bayesian inverse problem in the learned latent representation solves the same problem as the inversion in the original space with a pushforward prior.

(C3) The proposed methodology is naturally extended to the Bayesian estimation of other important quantities in the observation process, such as learning the observational noise.

(C4) We test the proposed methodology on: a steady-state heat problem, RANS flow around airfoils of varying geometries, damped Helmholtz resonance and source localization on car geometries. We compare our methods to supervised neural network-based methods and Graph Gaussian Processes where relevant.

(C5) We demonstrate the scalability of the method on a large terrain flow problem through a multi-GPU implementation, showing potential for training GABI foundation models.

Section 1.1 discusses the inference setup in detail; Section 1.2 goes over relevant related works; Section 2 describes the proposed methodology; Section 3 discusses the implementation of the proposed method; Section 4 tests and compares the proposed method; Appendices A, B, C, elaborate on proofs, additional numerical results and implementation details, relevant discussions on VAEs.

## 1.1 SETUP

Let $u_n \in \mathcal{U}_n := \mathcal{U}(\mathcal{M}_n; \mathbb{R}^{d_u})$ be a (possibly vector valued) function representing the full field behavior of a physical system on the bounded domain $\mathcal{M}_n \subset \mathbb{R}^d$. The space $\mathcal{U}$ is the relevant function space for the full field solution, e.g. a Sobolev space, and the geometric domain $\mathcal{M}_n$ may be a manifold with boundary or a boundaryless closed manifold. In this work, $u_n$ will be: heat on a surface, fluid flow in a domain, and resonance patterns on an object. Here, the index $n$ refers to the different physical systems which possibly have different geometries, forcing, and boundary conditions. In numerically representing these continuous systems, we obtain discretized node values $\mathbf{u}_n \in \mathsf{U}_n := \mathsf{U}(\mathsf{M}_n; \mathbb{R}^{d_u})$ which live on $\mathsf{M}_n = (\mathsf{V}_n, \mathcal{E}_n)$ where $\mathsf{V}_n = \{x_1, \ldots, x_{d_n}\} \subset \mathcal{M}_n$ is a set of vertices and $\mathcal{E}_n$ is a set of edges connecting the vertices. As such, we represent $\mathsf{M}_n$ as an undirected graph describing the physical geometries on which the physics is taking place, i.e., a computational mesh. We assume to have a dataset $\mathcal{D} = \{\mathbf{u}_n, \mathsf{M}_n\}_{n=1}^N$ [1]. This dataset can be constructed from simulation data, as is the case in this paper, or from full-field data from experiment such as through particle image velocimetry, digital image correlation, magnetic resonance, electrical impedance tomography, infrared thermography etc.

First, in the proposed methodology we learn a $\mathsf{M}_n$-dependent prior generative model for $\mathbf{u}_n$. Second, using this learned highly informative prior and *sparse noisy* observations, $\mathbf{y}_o \in \mathbb{R}^{d_o}$ [2], of a new

---

[1] A dataset with multiple full-fields per geometry $\mathsf{M}_n$, as $\{\{\mathbf{u}_{n,i}\}_{i=1}^{N_n}, \mathsf{M}_n\}_{n=1}^N$, can be viewed as a dataset $\{\mathbf{u}_k, \mathsf{M}_k\}_{k=1}^K$ indexed by a dummy variable $k$ such that $k$ maps to $(i, n)$.

[2] When discussing observations which are associated to a geometry not in the dataset $\mathcal{D}$ we change the index from $n$ to $o$ for clarity of exposition.

physical system on a new geometry, $\mathsf{M}_o$, we solve the Bayesian inverse problem to obtain the posterior distribution $p_{\mathbf{u}_o|\mathbf{y}_o}$. We assume to know the discretized geometry $\mathsf{M}_o$ associated to $\mathbf{y}_o$. In this paper, the observations and discretized solution field are related by:

$$\mathbf{y}_o = \mathbf{H}_o \mathbf{u}_o + \boldsymbol{\xi}_o, \quad \boldsymbol{\xi}_o \sim \mathcal{N}(\mathbf{0}, \sigma^2 \mathbf{I}_o). \tag{1}$$

We take $\mathbf{H}_o : \mathbb{R}^{d_u} \to \mathbb{R}^{d_o}$ to be a matrix of zeros and ones selecting the nodes to be observed. This is assumed to be a known observation operator corresponding to sensor placements. We note that many observation models can be considered with ease, such as surface integrals for computing drag coefficients, Radon transforms for tomography, as well as any number of noise models – additive or not. As our learned prior is entirely independent of the observation process, changes to the observation process do not necessitate retraining and can be incorporated "on-the-fly".

The proposed methodology works as follows:

- In the training stage, using a dataset of geometries and full-field solutions on these geometries, $\mathcal{D} = \{\mathbf{u}_n, \mathsf{M}_n\}_{n=1}^N$, we learn a graph-based autoencoder which embeds the joint distribution of solutions and geometries into a latent Gaussian distribution. This will act as our highly informative geometry-aware data-driven prior.
- In the application stage, we use this learned prior to solve inverse problems for data coming from new physical systems not in the dataset $\mathcal{D}$. That is, given *sparse and noisy* observations $\mathbf{y}_o$ of a solution $\mathbf{u}_o$ of a physical system with geometry $\mathsf{M}_o$, we use our previously learned prior to solve the Bayesian inverse problem for the unknown full-field $\mathbf{u}_o$. We recover a distribution over the solution $\mathbf{u}_o$ by only observing $\mathbf{y}_o$.

Using the proposed methodology, one can encode prior beliefs through geometry-dependent full-field solution datasets. Practitioners can then construct highly-informative priors from data across varying physical geometries. This prior is independent of the observation process for any particular inference task, hence this is a learn-once apply-everywhere approach.

## 1.2 RELATED WORKS

### 1.2.1 DIRECT MAP INVERSION

Many interesting approaches have been recently proposed which directly learn an amortized map from observable to solution/parameter. These can be subdivided into deterministic methods and probabilistic - often Bayesian - approaches. Most related to the setting considered in this paper is the work in Duthé et al. (2025). Here, the authors consider a graph-ML approach to learning a deterministic map from observable $\mathbf{y}_o$ to the underlying vector field $\mathbf{u}_o$. The supervised methodology used in the comparisons in this work draws inspiration from that prior work. Another strongly related methodology can be found in Arridge et al. (2019) although here the problem of changing geometry is not directly tackled. Other noteworthy works in this area include (Adler & Öktem, 2017; Dittmer et al., 2020). Interesting works look at probabilistic schemes for constructing conditional maps from observables to quantities of interest (Kaltenbach et al., 2023; Baptista et al., 2024; Hosseini et al., 2025). Other works in this area include (Adler et al., 2022; Ardizzone et al., 2018).

### 1.2.2 FORWARD AND INVERSE PROBLEMS ON GRAPHS

Graphs are handled in machine learning frameworks through geometric deep learning (Bronstein et al., 2021; Wu et al., 2022; Kipf, 2016). Several important works consider inverse problems where the quantity inferred lives on a graph description of the geometry. In Garcia Trillos & Sanz-Alonso (2018) the authors prove the convergence of the graph posterior to the continuum posterior subject to some conditions. In Povala et al. (2022) the adjacency structure of a mesh is leveraged to accelerate variational inference inversion. Important works consider solving the PDE forward problem through physics-informed graph machine learning (Seo & Liu, 2019; Valencia et al., 2025; Wandel et al., 2020). Used in our comparisons are linear inverse problems with Gaussian process priors, this leads to graph Gaussian process methods (Borovitskiy et al., 2021; Mostowsky et al., 2024). We find it important to mention works in the physics-informed literature where (assuming complete or partial knowledge on PDEs, boundary conditions, forcing etc.) one can use physics residuals to learn forward and inverse maps for PDEs (Raissi et al., 2019; Gao et al., 2022; Vadeboncoeur et al., 2023; Xu et al., 2023; Rixner & Koutsourelakis, 2021). However, in this work, we focus solely on methods which do not assume knowledge of the governing PDE, boundary conditions, or forcing and are entirely data-driven in their approach.

### 1.2.3 STATISTICAL AUTOENCODERS

Related to inverse problem are latent variable representations of graph data structures. In this category, we focus on the seminal work of the graph VAE (Kipf & Welling, 2016; Nazari et al., 2023). We refer the reader to Appendix C for a more elaborate discussion on the relationship between VAEs and our proposed framework. Related to this are the works (Mylonas et al., 2021; Liu et al., 2019). The proposed work centres around a class of autoencoders dubbed statistical autoencoders, of which there are many variants (Zhao et al., 2019; Kolouri et al., 2019; Turinici, 2021; Tolstikhin et al., 2018). We specifically focus on the MMD (Gretton et al., 2012) variant of the autoencoder. Many other variants could be considered. Another relevant class of autoencoders are conditional autoencoders (Sohn et al., 2015), we will make use of related ideas by conditioning encoder and decoder on geometry.

## 1.3 NOTATION

We denote by $\mathbb{P}_z$ a probability measure on the measurable space $(\mathcal{Z}, \mathcal{S})$, and by $p_z$ the associated density with respect to Lebesgue measure when it exists. Empirical measures are a mixture of Dirac measures. That is, for $D = \{x_i\}_{i=1}^M$, we have $\widehat{\mathbb{E}}_{x \in D}[\delta_x] = \frac{1}{M} \sum_{i=1}^M \delta_{x_i}$ where $\delta_{x_i}(A) = 1$ if $x_i \in A$, and $\delta_{x_i}(A) = 0$, if $x_i \notin A$. We will sometimes refer to such probability measures as the "densities" $p, q$ to lighten notation. Superscripts in parentheses denote realizations of random variables. For a function $g : \mathcal{Z} \to \mathcal{U}$, the pushforward $g_\# p_z$ means $g_\# \mathbb{P}_z(A) = \mathbb{P}_z(g^{-1}(A))$ for all measurable sets $A$. We note that a sample $u^{(n)} \sim g_\# p_z$ is obtained as $u^{(n)} = g(z^{(n)}), z^{(n)} \sim p_z$.

## 2 METHODOLOGY

We now describe the proposed methodology in detail. In Section 2.1 we carefully lay out the central result allowing for the Bayesian treatment of the inference task; that is, the equivalence between the Bayesian posterior of a pushforward prior (our generative model), and the Bayesian inverse problem in the implied latent space of the pushforward prior. Following from this result, in Section 2.2 we describe our two-step procedure: first, learn a generative model to be used as a Geometry-conditioned Bayesian prior; second, observe sparse data for the inverse problem and solve the Bayesian inverse problem in the latent space to push this posterior back out to the desired physical field living on the geometry of interest. In Section 2.3 we show how to extend our methodology to estimate the noise in the observations.

### 2.1 POSTERIOR FROM A PUSHFORWARD PRIOR

We present the lemma motivating our methodology; the equivalence between the Bayesian posterior with a pushforward latent prior and the pushforward of the Bayesian posterior over the latent space.

**Lemma 2.1.** *Let $(\mathcal{U}, \mathcal{F}, \mathbb{P}_u)$, and $(\mathcal{Z}, \mathcal{S}, \mathbb{P}_z)$ be measure spaces and $g : \mathcal{Z} \to \mathcal{U}$ be $(\mathcal{S}, \mathcal{F})$-measurable. Furthermore, let $\mathbb{P}_{u|y}$ be the Bayesian posterior on $\mathcal{U}$ with likelihood proportional to $\exp(-\Phi(u; y))$ and prior $\mathbb{P}_u := g_\# \mathbb{P}_z$. We find*

$$\mathbb{P}_{u|y} = g_\# \mathbb{P}_{z|y},$$

*where $d\mathbb{P}_{z|y}(z) = 1/Z(y) \exp(-\Phi(g(z); y)) d\mathbb{P}_z(z)$ is the Bayesian posterior on $\mathcal{Z}$.*

Stated simply, in Bayesian inference, when a prior on $\mathcal{U}$ is specified as the pushforward under $g$ (Subsection 1.3) of a latent prior on a space $\mathcal{Z}$, two distributions are equivalent: i) the Bayesian posterior on $\mathcal{U}$, and ii) the pushforward under $g$ of the posterior associated to the latent prior on $\mathcal{Z}$. This result will be explicitly related to the geometric inference setting in Theorem 2.2, showing that with a latent prior constructed via a geometry-aware encoder/decoder, we can solve inverse problems on $\mathcal{U}$ with a single common latent representation relevant to any number of in-distribution geometries. All proofs can be found in Appendix A.

### 2.2 GEOMETRIC AUTOENCODER PRIORS FOR BAYESIAN INVERSION (GABI)

We describe the two-step process of learning a geometry-aware prior and how to use this to solve inverse problems.

**Step 1: Learn Geometric Autoencoder Prior.** Let $\mathbf{z} \in \mathsf{Z} = \mathbb{R}^{d_z}$ be a latent vector. We pose the $\mathsf{M}_n$-dependent maps

$$E_n^\theta : \mathsf{U}_n \to \mathsf{Z}, \quad \text{where} \quad E_n^\theta(\mathbf{u}_n) := E^\theta(\mathbf{u}_n; \mathsf{M}_n); \tag{2a}$$

$$D_n^\psi : \mathsf{Z} \to \mathsf{U}_n, \quad \text{where} \quad D_n^\psi(\mathbf{z}) := D^\psi(\mathbf{z}; \mathsf{M}_n). \tag{2b}$$

Both $E^\theta$ (the encoder) and $D^\psi$ (the decoder) are graph neural networks with parameter sets $\theta, \psi$ respectively. These two maps imply a geometry conditioned autoencoder. We use $\widehat{\mathbb{E}}_\mathcal{D}[\,\cdot\,]$ as a shorthand for $\frac{1}{N} \sum_{(\mathbf{u}_n, \mathsf{M}_n) \in \mathcal{D}}[\,\cdot\,]$. Set the parameters of the auto-encoder to

$$\theta^\star, \psi^\star = \underset{\theta, \psi}{\arg\min} \; \widehat{\mathbb{E}}_\mathcal{D}[\|\mathbf{u}_n - (D_n^\psi \circ E_n^\theta)(\mathbf{u}_n)\|_2^2] + \mathsf{d}(p_\mathbf{z}^\theta, q_\mathbf{z}), \tag{3}$$

where $p_\mathbf{z}^\theta = \widehat{\mathbb{E}}_\mathcal{D}[\delta_{E_n^\theta(\mathbf{u}_n)}]$ is the empirical distribution of encoded solutions, $q_\mathbf{z} = \mathcal{N}(\mathbf{0}, \mathbf{I})$, $\arg\min$ is to be approximated with gradient descent, and $\mathsf{d} : \mathcal{P}(\mathsf{Z}) \times \mathcal{P}(\mathsf{Z}) \to [0, \infty)$ may be any suitable divergence that is well-posed for comparing empirical measures: maximum mean discrepancy (MMD), energy distance (ED), Wasserstein, or Sliced-Wasserstein distances etc. In this work, we focus on MMD.

**Step 2: Observe and Sample Posterior.** For new data $\mathbf{y}_o$ associated to a geometry $\mathsf{M}_o$, we re-write the data generating model in (1) in terms of a decoded latent variable as

$$\mathbf{y}_o = \mathbf{H}_o D_o^\psi(\mathbf{z}) + \boldsymbol{\xi}_o, \quad \boldsymbol{\xi}_o \sim \mathcal{N}(\mathbf{0}, \sigma^2 \mathbf{I}_o), \tag{4}$$

where $D_o^\psi = D^\psi(\,\cdot\,; \mathsf{M}_o)$. We specify our prior distribution over $\mathbf{u}_o$, denoted $p_{\mathbf{u}_o}$, using the trained decoder; and specify the likelihood, denoted $p_{\mathbf{y}_o|\mathbf{z}}^\psi$, associated to this decoded observation model (4)

$$p_{\mathbf{u}_o}^\psi := D_{o\,\#}^\psi q_\mathbf{z}; \qquad p_{\mathbf{y}_o|\mathbf{z}}^\psi = \mathcal{N}(\mathbf{H}_o D_o^\psi(\mathbf{z}), \sigma^2 \mathbf{I}_o). \tag{5}$$

This likelihood along with prior $q_\mathbf{z}$ imply a posterior

$$p_{\mathbf{z}|\mathbf{y}_o}^\psi(\mathbf{z}) = \frac{p_{\mathbf{y}_o|\mathbf{z}}^\psi(\mathbf{y}_o)\, q_\mathbf{z}(\mathbf{z})}{p_{\mathbf{y}_o}(\mathbf{y}_o)}. \tag{6}$$

**Theorem 2.2.** *Assuming the relevant measurability of $D_o^\psi$, the posterior*

$$p_{\mathbf{u}_o|\mathbf{y}_o}^\psi = D_{o\,\#}^\psi p_{\mathbf{z}|\mathbf{y}_o}. \tag{7}$$

The posterior (7) is sampled by first sampling from $p_{\mathbf{z}|\mathbf{y}_o}^\psi$ in (6) and decoding with $D_o^\psi$. Hence, by first solving the inverse problem in the latent space, we can use Theorem 2.2 to easily obtain posterior samples in the decoded solution space for a given geometry. This allows us to learn a unified prior over the solution field as a generative model *over varying geometries* and use it to solve Bayesian inverse problems with a highly informative learned prior on new geometries.

## 2.3 Observational Noise Estimation

Beyond the benefits of quantifying uncertainty related to model predictions, the Bayesian paradigm offers a principled framework for estimating other aspects of the data-generating model. We highlight that such estimations are completely independent of the training of the autoencoder prior – this is not the case for direct regression methods which need to have all observation variables incorporated at training time. We now consider the common task of jointly estimating the observational noise standard deviation $\sigma_o$ (now indexed by $o$ as it may vary from one system to another), together with the field of interest $\mathbf{u}_o$. This is achieved through the latent joint posterior

$$p_{\mathbf{z}, \sigma_o|\mathbf{y}_o}^\psi(\mathbf{z}, \sigma_o) = \frac{p_{\mathbf{y}_o|\mathbf{z}, \sigma_o}^\psi(\mathbf{y}_o) p_{\mathbf{z}, \sigma_o}(\mathbf{z}, \sigma_o)}{p_{\mathbf{y}_o}(\mathbf{y}_o)}. \tag{8}$$

where $p_{\mathbf{z}, \sigma_o}(\mathbf{z}, \sigma_o) = q_\mathbf{z}(\mathbf{z}) p_{\sigma_o}(\sigma_o)$ for a specified prior $p_{\sigma_o}(\sigma_o)$.

**Corollary 2.3.** *We can sample the joint posterior on the solution fields through the pushforward* $p_{\mathbf{u}_o, \sigma_o|\mathbf{y}_o}^\psi = (D_o^\psi \otimes \mathrm{Id}.)_\# \, p_{\mathbf{z}, \sigma_o|\mathbf{y}_o}^\psi$, *where* $\mathrm{Id}.$ *is the identity map.*

---

**Algorithm 1:** Inversion with GABI-ABC

---

1: Specify the observation vector $\mathbf{y}_o$, discretized geometry $\mathsf{M}_o$, total sample budget $N_s$, number of accepted samples $N_a$.
2: Train $E^\theta$, $D^\psi$ through (3) on $\mathcal{D}$.
3: **for** [Batched/Parallelized] $i = 1 : N_s$ **do**
4:     Sample $\mathbf{z}^{(i)} \sim q_{\mathbf{z}}$
5:     Decode $\mathbf{u}_o'^{(i)} = D^\psi(\mathbf{z}^{(i)}; \mathsf{M}_o)$
6:     Apply observation process with noise: $\mathbf{y}_o'^{(i)} = \mathbf{H}_o \mathbf{u}_o'^{(i)} + \boldsymbol{\xi}_o'^{(i)}; \quad \boldsymbol{\xi}_o'^{(i)} \sim \mathcal{N}(\mathbf{0}, \sigma^2 \mathbf{I}_o)$.
7:     Compute $r_i = \|\mathbf{y}_o - \mathbf{y}_o'^{(i)}\|_2$.
8: **return** return $N_a$ samples of $\mathbf{u}_o^{(1:N_s)}$ with least residual $r_i$.

---

## 3 IMPLEMENTATION

**Neural Networks:** We make use of meshes and graph neural networks to describe and manipulate geometric information. The proposed methodology is agnostic to any particular architecture, however we need the following characteristics: *Geometry-aware architecture and data-structure; Mapping to and from fixed-dimensional vectors; Non-locality*. We expand on these points in B. For the heat, airfoil, and car problems we make use of a Graph Convolutional Network with interleaved nonlocal averaging layers (Lanthaler et al., 2024). For the terrain flow problem we use a GEN (Li et al., 2020) GNN architecture. We detail these architectures in Appendix B.1. We also test a Transformer on the heat problem in Appendix B.3.3.

**Sampling:** To realize GABI, we first train the autoencoder; then, we sample from the corresponding latent posterior as in (6) and decode the resulting latent samples to obtain samples from the posterior for a given geometry. We note two alternatives for sampling from the posterior: MCMC, and ABC sampling. We find ABC sampling to be advantageous for this task due to: (i) the massive parallelization capabilities of neural networks as opposed to the sequential nature of MCMC; (ii) the specification of a well-tailored prior distribution – the prior is not unnecessarily vague leading to good overlap between the prior and posterior; (iii) the low dimensionality of the observational data – removing the need to use possibly problematic summary statistics. In Algorithm 1 we outline the implementation of the GABI-ABC variant. For MCMC sampling, we use the NUTS algorithm (Hoffman et al., 2014).

**Notes on Alternative Approaches: Physics Driven Inversion.** These require additional knowledge aside from access to a representative dataset and information specific to each system we wish to query such as the governing PDE, boundary conditions, forcing etc. **VAEs.** We refer the reader to Appendix C for an in-depth discussion on the unsuitability of VAEs for the task we are solving, both in terms of the overall methodology and in terms of the autoencoder used within the proposed methodology. **Supervised learning/Direct Map Inversion.** One can choose to learn a supervised learning direct mapping $(\mathbf{y}_n, \mathsf{M}_n) \mapsto \mathbf{u}_n$; this approach is the main point of comparison with our method. The significant disadvantage of this approach is that the observation process (in this paper $\mathbf{y}_n = \mathbf{H}_n \mathbf{u}_n + \boldsymbol{\xi}_n$) must be known at training time and kept the same at inference time (or kept the same in a distributional sense, as explained in the comparisons). However, in practice, the observation process for different physical systems can vary enormously as we measure different quantities such as moments, local averages, summary coefficients, have a variable number of observations, or vary the type and magnitude of noise etc. In contrast to this, GABI is independent of the observation process at training time and only requires this information at test time. Thus, GABI offers a train-once-use-everywhere type foundation model, hugely advantageous for practical/industry uptake of ML models. **Gaussian Process Regression.** For an inverse problem over fields, a natural choice of prior is a Gaussian process prior. As the observation process $\mathbf{H}_o$ is linear, the classical Bayesian posterior for this problem is precisely a Gaussian process regression (GPs). We perform hyper-parameter optimization of this GP, which relates it to common usage of GPs in the context of ML. The GPs are defined directly on the graphs (Borovitskiy et al., 2021). **Bayesian NNs.** The problem of specifying informative priors over varying geometries persists and inherits the problems of direct maps discussed above.

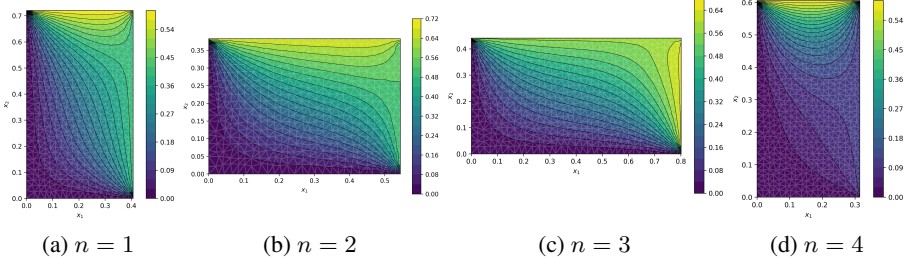

| (a) $n = 1$ | (b) $n = 2$ | (c) $n = 3$ | (d) $n = 4$ |

Figure 1: Four out of 1k geometries and solutions in dataset for the steady-state heat problem.

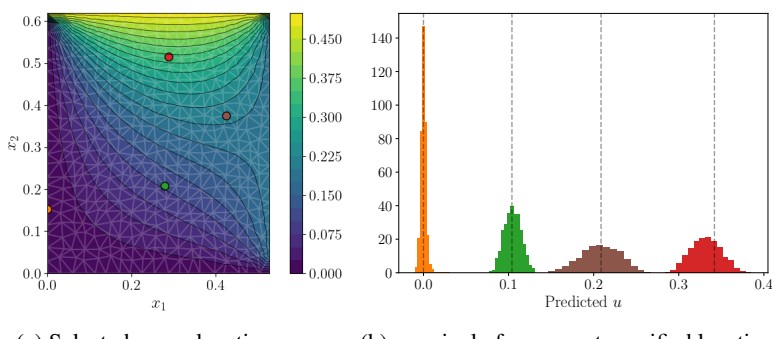

(a) Selected query locations      (b) marginal of $p_{\mathbf{u}_n|\mathbf{y}_n}$ at specified locations

Figure 2: (a) Four selected query locations for the sampled predictive solutions given data, the full field estimations are in Figure 6. (b) The corresponding histograms for the predictive posterior at specified query locations; the dashed black line indicates the ground truth for each histogram.

## 4   NUMERICS

In this section we present numerical results for steady-state heat on rectangles, flow fields around airfoils, damped resonance and source localization on car bodies, and terrain flows. We compare our method to direct regression (Direct Map) using graph NNs, and Gaussian process regression with Matérn kernels $1/2, 3/2$, and radial basis function (RBF). In all experiments we report the wall-clock training time (Train) on Nvidia RTX 4090 GPUs (the terrain problem is multi-GPU), and the mean per-geometry prediction time (Pred.). We also note that in using ABC for sampling from the posterior we do not need to compute the likelihood function. This may have strong advantages in application where the observation process yields intractable likelihoods. In using GABI with MCMC, as in the comparisons, we do compute the likelihood. In training the latent prior for GABI, the main way we have of knowing if the autoencoder networks perform adequately for the inference task, is by assessing the quality of reconstruction for a holdout set, as well as how close the embed holdout set is to being Gaussian in the latent space, which is assessed qualitatively and quantitatively.

### 4.1   RECTANGLES – STEADY-STATE HEAT

In this section, we study the homogeneous steady-state heat equation on rectangular domains with varying width, height, and boundary conditions. Thus, for a given rectangle $\mathcal{M}_n$, boundary condition $h_n$, any $u_n$ satisfies the steady-state heat equation. We take $\mathcal{M}_n \subset \mathbb{R}^2$ to be $\mathcal{M}_n = \{x \in \mathbb{R}^2 : 0 < x_1 < l_n, 0 < x_2 < w_n\}$. The domain $\mathcal{M}_n$ is defined by its length and width $(l_n, w_n)$ drawn from `uniform`$((0.1, 0.1), (1, 1))$; we specify $h_n$ to be zero on the bottom and left side, and drawn from `uniform`$((0.1, 0), (1, 1))$ on the top and right. The solution to (9) depends on $\mathcal{M}_n, h_n$. We generate 1k solutions to be in the training dataset. In Figure 1 we show example geometries and solutions in the dataset. Once the model is trained on this data (the loss is shown in B.3) we select a new observation vector and geometry $\mathsf{M}_o, \mathbf{y}_o$ and use ABC sampling as in Algorithm 1. Figure 2 (b) shows 4 coloured histograms corresponding to the marginal distribution of a decoded posterior

Table 1: Comparison on Heat Equation in Rectangular Domain

| Method | Field | MAE | % 1 std | % 2 std | Train | Pred. |
|---|---|---|---|---|---|---|
| **2D Heat Rect. Domain** | | | | | | |
| GABI-ABC | $(u)$ | $1.58 \cdot 10^{-2} \pm 1.36 \cdot 10^{-2}$ | 80.91% | 95.59% | 2.62hr | 0.908s |
| GABI-NUTS | $(u)$ | $1.11 \cdot 10^{-2} \pm 1.04 \cdot 10^{-2}$ | 66.66% | 96.18% | ” | 410.31s |
| Direct Map | $(u)$ | $1.25 \cdot 10^{-2} \pm 1.02 \cdot 10^{-2}$ | – | – | 1.47hr | $0.0029s$ |
| GP (M 1/2) | $(u)$ | $8.46 \cdot 10^{-2} \pm 4.18 \cdot 10^{-2}$ | 66.36% | 89.45% | – | 0.65s |
| GP (M 3/2) | $(u)$ | $8.06 \cdot 10^{-2} \pm 4.85 \cdot 10^{-2}$ | 65.80% | 87.22% | – | 0.64s |
| GP (RBF) | $(u)$ | $1.39 \cdot 10^{-1} \pm 6.66 \cdot 10^{-2}$ | 14.88% | 29.36% | – | 0.64s |

Table 2: Comparison on Heat Equation in Rectangular Domain – Unknown Noise

| Method | Field/QoI | MAE | % 1 std | % 2 std | Train | Pred. |
|---|---|---|---|---|---|---|
| GABI-ABC | $(u)$ | $2.09 \cdot 10^{-2} \pm 1.62 \cdot 10^{-2}$ | 75.74% | 94.76% | 2.62hr | 0.904s |
| Direct Map | $(u)$ | $2.13 \cdot 10^{-2} \pm 2.27 \cdot 10^{-2}$ | – | – | 1.46hr | 0.0030s |
| GP (M 1/2) | $(u)$ | $7.73 \cdot 10^{-2} \pm 5.18 \cdot 10^{-2}$ | 64.87% | 87.50% | – | 0.615s |
| GP (M 3/2) | $(u)$ | $7.56 \cdot 10^{-2} \pm 5.56 \cdot 10^{-2}$ | 51.58% | 76.94% | – | 0.686s |
| GP (RBF) | $(u)$ | $1.40 \cdot 10^{-1} \pm 6.94 \cdot 10^{-2}$ | 18.03% | 36.69% | – | 0.694s |
| GABI-ABC | $(\sigma)$ | $7.96 \cdot 10^{-1} \pm 5.74 \cdot 10^{-1}$ | 42.10% | 69.11% | – | “ “ |
| Direct Map | $(\sigma)$ | – | – | – | – | – |
| GP (M 1/2) | $(\sigma)$ | $1.08 \cdot 10^{0} \pm 1.03 \cdot 10^{0}$ | – | – | – | “ “ |
| GP (M 3/2) | $(\sigma)$ | $1.10 \cdot 10^{0} \pm 1.47 \cdot 10^{0}$ | – | – | – | “ “ |
| GP (RBF) | $(\sigma)$ | $2.01 \cdot 10^{0} \pm 3.61 \cdot 10^{0}$ | – | – | – | “ “ |

(based on 10 sparse observations in the vector $\mathbf{y}_o$) at the 4 queried locations on Figure 2 (a). As the orange dot is close to a boundary with value 0 in the entire dataset (4 representative samples from the 1k geometry dataset can be seen in Figure 1), GABI presents very small uncertainty in this area. As we query points in areas where there is more variability of the solution in the dataset, GABI presents wider uncertainty. In all cases the mean of the GABI posterior is very close to the ground truth (vertical dashed line). This exemplifies the balance GABI strikes between sparse data and informative data-driven priors. **Comparison – Known Noise.** In Table 1 we compare two sampling variants of GABI with a direct regression map as well as Gaussian Process Kriging with various kernels[3]; M stands for Matérn. In this setup, we keep the number of observations at 10 randomly selected locations and the noise standard deviation at $10^{-2}$. **Comparison – Unknown Noise.** In Table 2 we test the noise estimation capabilities of GABI. We have the noise standard deviation be random and drawn from a shifted log-normal distribution as $\sigma_n = \exp(\varepsilon_n - 4) + 10^{-3}; \varepsilon_n \sim \mathcal{N}(0, I)$. We estimate the noise for each data $\mathbf{y}_o$ with the GABI approach outlined in 2.3.

## 4.2 AIRFOILS – FLOW FIELD

In this section, we test the proposed methodology on the full field reconstruction or airflow around an airfoil from a dataset of Reynolds averaged Navier-Stokes simulations. We use the setup described in Duthé et al. (2025). Here, the geometric autoencoder is trained on the pressure and 2D velocity field of 1k airfoil geometries with various inflow conditions. Once trained, the observation vector $\mathbf{y}_o$ is taken to be noisy pressure observations at a small number of nodes on the surface of the airfoil, emulating a physical scenario when such pressure sensors may be attached to a wing/blade. The Bayesian inversion task is to reconstruct the pressure and velocity field around the airfoil, a task far too ill-posed without a highly informative prior. To compile the results in Table 3 we test GABI-ABC against the Direct Map with the same architecture type – we randomize the number of observation locations to be drawn between 5 and 50 observations with probability inversely proportional to the number of observation locations. For the supervised Direct Map approach the distribution of the number of observation must be known at train time, this is not the case for GABI whose training is

---

[3]GABI-NUTS was tested for $10^{2}$ runs not $10^{3}$ due to prediction runtime.

Table 3: Comparison on Airfoil

| Method | Field | MAE | % 1 std | % 2 std | Train | Pred. |
|--------|-------|-----|---------|---------|-------|-------|
| GABI-ABC | $(p)$ | $6.92 \cdot 10^{-2} \pm 4.39 \cdot 10^{-2}$ | 78.77% | 97.28% | 8.61hr | 34.76s |
| Direct Map | $(p)$ | $5.35 \cdot 10^{-2} \pm 3.07 \cdot 10^{-2}$ | – | – | 4.68hr | 0.0040s |
| GABI-ABC | $(v_x)$ | $1.31 \cdot 10^{-1} \pm 3.62 \cdot 10^{-2}$ | 80.36% | 97.28% | " " | " " |
| Direct Map | $(v_x)$ | $9.30 \cdot 10^{-2} \pm 1.93 \cdot 10^{-2}$ | – | – | " | " |
| GABI-ABC | $(v_y)$ | $3.94 \cdot 10^{-2} \pm 2.20 \cdot 10^{-2}$ | 75.87% | 96.08% | " " | " " |
| Direct Map | $(v_y)$ | $3.24 \cdot 10^{-2} \pm 1.12 \cdot 10^{-2}$ | – | – | " | " |

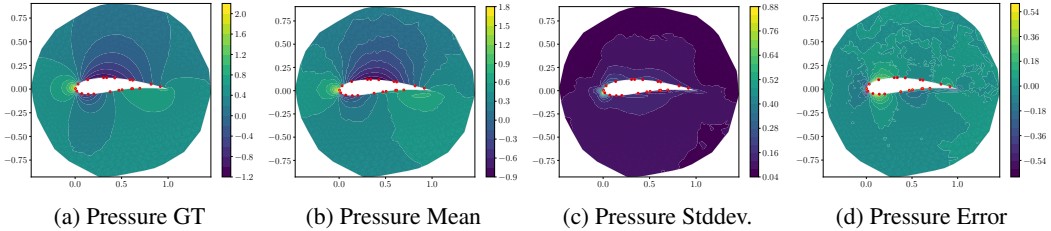

(a) Pressure GT  (b) Pressure Mean  (c) Pressure Stddev.  (d) Pressure Error

Figure 3: Comparison of ground truth (GT), inferred mean, error, and standard deviation for pressure ($p$). The red dots correspond to the observation location for the pressure. Full results including reconstruction of vertical and horizontal velocity fields are in Figure 11

Table 4: Comparison on Car Resonance – Full Field Reconstruction and Source Localization

| Method | Field | MAE | % 1 std | % 2 std | Train | Pred. |
|--------|-------|-----|---------|---------|-------|-------|
| GABI-ABC | $(u)$ | $2.32 \cdot 10^{-1} \pm 7.21 \cdot 10^{-2}$ | 75.48% | 96.88% | 2.29hr | 18.14s |
| Direct Map | $(u)$ | $6.03 \cdot 10^{-1} \pm 1.46 \cdot 10^{-1}$ | – | – | 1.22hr | 0.0032s |
| GABI-ABC | $(f)$ | $2.26 \cdot 10^{-3} \pm 4.36 \cdot 10^{-4}$ | 80.55% | 97.49% | " | " |
| Direct Map | $(f)$ | $2.46 \cdot 10^{-1} \pm 8.76 \cdot 10^{-2}$ | – | – | " " | " " |

independent of the observation process. Figure 3 shows a select number of results. For a discussion on out-of-distribution inference we refer the reader to Appendix B.4.2.

## 4.3 CAR BODY – ACOUSTIC VIBRATION AND SOURCE LOCALIZATION

Here, we study an acoustic resonance problem with a source localization task. We have a collection of car body geometries which undergo damped vibration due to a localized Gaussian bump forcing present in the first $1/5$ of the vehicle (where the engine is), the inversion task is to reconstruct the full field acoustic vibration amplitude, $u$, as well as the full field forcing, $f$, on the surface of the vehicle from a small number of sparse noisy measurements of the vibration amplitude. We do not observe the forcing field. The car geometries are taken from (Umetani & Bickel, 2018). We train the model on 500 car geometries, the number of observation locations is randomized as in Section 4.2. The numerical results are in Table 4 and we show a reconstruction result in Figure 4.

## 4.4 TERRAIN – FLOW FIELD

To demonstrate the scalability of our approach, we apply it to the reconstruction of flow fields over complex terrain. Here, we use a large dataset (127GB) of more than 5k RANS simulations, produced by Achermann et al. (2024). From the original $64^3$ grids, we extract 8-voxel-thick, terrain-conforming layers, resulting in $64 \times 64 \times 8$ subdomains. Each subdomain is then converted into a graph, where voxels become nodes connected to their adjacent neighbors. We scale both the training and the inference methods with multi-GPU pipelines. Using four RTX4090 GPUs, training takes 52hr, while inference takes 480s. Figure 5 shows the GABI-ABC pressure and velocity results.

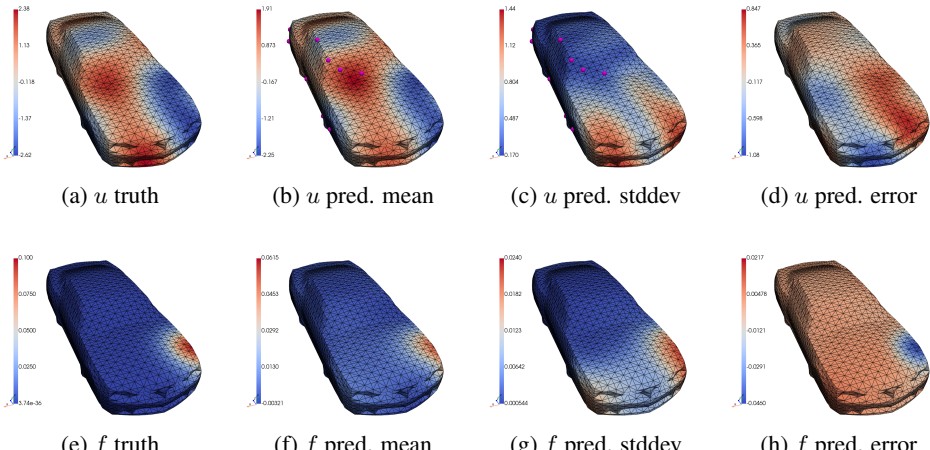

(a) $u$ truth  (b) $u$ pred. mean  (c) $u$ pred. stddev  (d) $u$ pred. error

(e) $f$ truth  (f) $f$ pred. mean  (g) $f$ pred. stddev  (h) $f$ pred. error

Figure 4: Ground truth, inferred mean, stddev., and error for amplitude $u$, and forcing $f$. In magenta are the observation locations.

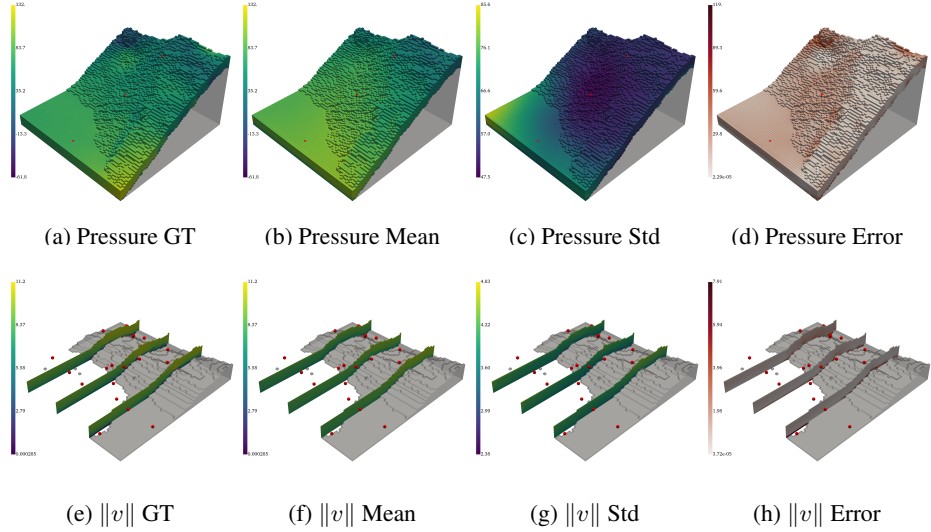

(a) Pressure GT  (b) Pressure Mean  (c) Pressure Std  (d) Pressure Error

(e) $\|v\|$ GT  (f) $\|v\|$ Mean  (g) $\|v\|$ Std  (h) $\|v\|$ Error

Figure 5: Ground truth, inferred mean, error, and standard deviation for pressure and the magnitude of the velocity vector, ($\|v\|$) across two terrains. The red dots correspond to the observation locations.

## 5 CONCLUSION

We propose a methodology for learning highly informative priors for solving Bayesian inverse problems across varying geometries. The scheme is realized through a geometric autoencoder which learns informative latent representations allowing for sharing of information across geometries. Once trained this foundation model can be used with any observation process to perform Bayesian full field reconstruction over new geometries. A creative use of ABC sampling leads to an efficient GPU implementation. The method achieves similar predictive accuracy to supervised direct map methods, all while outputting a full Bayesian posterior over the fields of interest and being far more flexible due to the model being independent of the observation process. The method is tested on a wide range of physical setups and geometries.

## ACKNOWLEDGEMENTS

AV is supported through the EPSRC ROSEHIPS grant [EP/W005816/1]. MG is supported by a Royal Academy of Engineering Research Chair and EPSRC grants [EP/X037770/1, EP/Y028805/1, EP/W005816/1, EP/V056522/1, EP/V056441/1, EP/T000414/1 and EP/R034710/1]. GD and EC are supported by the BRIDGE Discovery Program of the Swiss National Science Foundation and Innosuisse (Grant No. 40B2-0_187087), as well as the French-Swiss project MISTERY funded by the French National Research Agency (ANR PRCI Grant No. 266157) and the Swiss National Science Foundation (Grant No. 200021L_212718).

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

# A  PROOFS

## A.1  PROOF OF LEMMA 2.1

*Proof.* We remember

$$Z(y) = \int_{\mathcal{U}} \exp\left(-\Phi(u;y)\right) \mathrm{d}\mathbb{P}_u, \qquad \mathbb{P}_u := g_{\#}\mathbb{P}_z.$$

Thus, assuming the relevant measurabilities,

$$\begin{aligned} Z(y) &= \int_{\mathcal{U}} \exp\left(-\Phi(u;y)\right) \mathrm{d}(g_{\#}\mathbb{P}_z) \\ &= \int_{\mathcal{Z}} \exp\left(-\Phi(g(u);y)\right) \mathrm{d}\mathbb{P}_z, \end{aligned}$$

where the second equality is given by Theorem 3.6.1 in Bogachev & Ruas (2007). Applying the same equality, $\forall A \in \mathcal{F}$,

$$\begin{aligned} \mathbb{P}_{u|y}(A) &= \frac{1}{Z(y)} \int_A \exp\left(-\Phi(u;y)\right) \mathrm{d}(g_{\#}\mathbb{P}_z) \\ &= \frac{1}{Z(y)} \int_{\mathcal{U}} \mathbb{1}_A(u) \exp\left(-\Phi(u;y)\right) \mathrm{d}(g_{\#}\mathbb{P}_z) \\ &= \frac{1}{Z(y)} \int_{\mathcal{Z}} \mathbb{1}_A(g(z)) \exp\left(-\Phi(g(z);y)\right) \mathrm{d}\mathbb{P}_z \\ &= \frac{1}{Z(y)} \int_{\mathcal{Z}} \mathbb{1}_{g^{-1}(A)}(z) \exp\left(-\Phi(g(z);y)\right) \mathrm{d}\mathbb{P}_z \\ &= \frac{1}{Z(y)} \int_{g^{-1}(A)} \exp\left(-\Phi(g(z);y)\right) \mathrm{d}\mathbb{P}_z \\ &= \mathbb{P}_{z|y}(g^{-1}(A)) \\ &= (g_{\#}\mathbb{P}_{z|y})(A), \end{aligned}$$

where $\mathbb{1}_A$ is the indicator function. Hence, $\mathbb{P}_{u|y} = g_{\#}\mathbb{P}_{z|y}$. $\qquad\square$

## A.2 Proof of Theorem 2.2

*Proof.* Using Lemma 2.1, replacing $g$ with $D_o^\psi$, identifying $z, u, y$ with $\mathbf{z}, \mathbf{u}_o, \mathbf{y}_o$, we obtain the desired posterior. $\square$

## A.3 Proof of Corollary 2.3

*Proof.* This follows directly from Lemma 2.1 and Theorem 2.2 by identifying $z$ with $(\mathbf{z}, \sigma_o)$, $\mathbb{P}_z$ with $(\mathbb{P}_\mathbf{z} \otimes \mathbb{P}_{\sigma_o})$, $g$ with $(D_o^\psi \otimes \mathrm{Id}.)$, changing the potential function to $\Phi(\mathbf{u}_o, \sigma_o; \mathbf{y}_o) = \frac{1}{2\sigma_o^2} \|\mathbf{y}_o - \mathbf{H}_o \mathbf{u}_o\|_\mathbf{2}^\mathbf{2}$, and finally noting $\mathrm{Id}._\# \mathbb{P}_{\sigma_o} = \mathbb{P}_{\sigma_o}$. $\square$

# B Additional Numerical Results and Details

The proposed methodology is architecture-agnostic, however for an architecture to be appropriate for the task we require a few particular qualities:

- ***Geometry-aware architecture and data-structure.*** As the main premise of this work is to share information about physical fields which depend on the geometry of the problem to which they relate, it is essential that the data-structure reflect this geometric information, that the neural network architecture be effective at processing it, and that both naturally accommodate inputs of varying size and topology.
- ***Mapping to and from fixed-dimensional vectors.*** This framework relies on learning latent representations between solution to PDEs and the geometries on which these are defined. Hence, one needs to map variable-sized geometric inputs to and from fixed-dimensional latent spaces.
- ***Non-locality.*** All physical processes in this paper behave in non-local ways, the solution field at one point in the domain affects the solution at all other points in the domain. Hence, when encoding solution fields for a given geometry, the encoder/decoder should reflect this non-locality.

We use three architectures, a Graph Convolutional Network (explained in B.1), a Generalized Aggregation Network (explained in B.2), and a transformer (explained and tested in B.3.3).

## B.1 Graph Convolutional Network

We use graph convolutional layers (Kipf, 2016) in the examples heat on rectangles, RANS around airfoil, Helmholtz on car. To induce non-locality of mappings across channels, $c$, at each layer we set $\mathrm{GCN}_l : \mathsf{M}_n \times \mathbb{R}^{2c} \to \mathsf{M}_n \times \mathbb{R}^c$ and the output is concatenated with a graph level channel-wise average which is expanded to populate each node of the graph. The edge attributes are the distance between nodes. The input node values always contain the node coordinates, and, for the encoder, also the data solution at the node.

**In GABI:** In the encoder $E^\theta$ we map attributed graphs to latent vectors of fixed dimensions. The decoder maps a pairing between a latent variable and a graph and populates the node attributes on the input graph. Hence the last layer in a channel-wise averaging across the graph and the channel dimension is taken to be the latent dimension. The decoder $D^\psi$ takes as argument a graph where the node attributes are only the coordinates, hence only contains geometric information, and we expand the latent sample $\mathbf{z}$ to decode to populate all nodes with a channel dimension equal to the latent dimension.

**In Direct Map:** For the direct map method we use the same GCN layers as explained above. However now we must map observation vectors $\mathbf{y}_n$ and geometry $\mathsf{M}_n$ directly to the solution on the geometry, $\mathbf{u}_n$. To do this, we make the input graph node attributes the coordinates of the nodes as per the previous section, the observed nodal values $\mathbf{y}_n$ (where other values are 0), and a one-hot encoding indicates where this node is observed or not.

**GP Regression:** We use a graph Matérn Gaussian process (Mostowsky et al., 2024) with maximum marginal likelihood estimation of the hyperparameters.

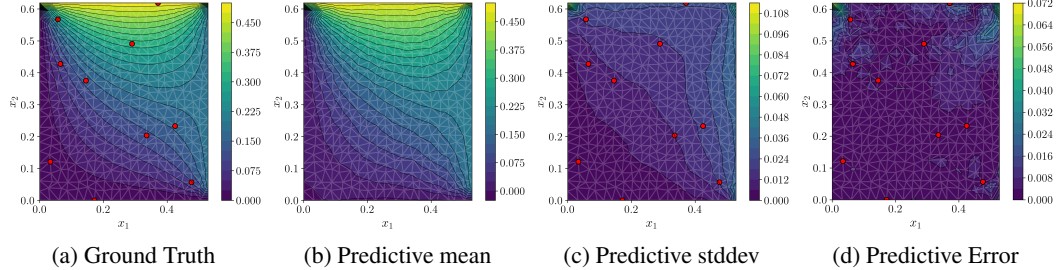

(a) Ground Truth  (b) Predictive mean  (c) Predictive stddev  (d) Predictive Error

Figure 6: For the heat setup, we overlay in red the observation locations. (a) ground truth (b) decoded mode of predictive posterior. (c) empirical standard deviation of decoded posterior samples. (d) error between decoded mode and ground truth.

The rectangle, airfoil, and car examples are run on a single RTX4090 GPU. The terrain example is run in a multi-GPU manner.

## B.2 Generalized Aggregation Network (GEN)

For the graph-based autoencoder applied to the flow over complex terrain, we replace standard GCN layers with Generalized Aggregation Network (GEN) layers (Li et al., 2020). This change allows us to add more complex geometrical information to the edge features (relative coordinates, similarly to (Pfaff et al., 2020)), as GEN layers, unlike GCNs, can process multidimensional edge features. This message-passing formulation also uses a softmax-based message aggregation scheme, which allows the model to dynamically weigh the importance of messages from neighboring nodes. Importantly, we maintain the same non-local pooling operation for each layer as for the GCN-based approach.

## B.3 Heat

In this section, we add numerical information and results for the steady-state heat problem. The equation used to generate the dataset is

$$\Delta u(x) = 0, \qquad x \in \mathcal{M}_n, \tag{9a}$$
$$u(x) = h_n(x), \quad x \in \partial \mathcal{M}_n. \tag{9b}$$

where $h_n$ is described in Section 4.1. We use a finite element solver over an unstructured triangular mesh to solve the systems to be in the dataset.

### B.3.1 Implementation Details

**Train:** We use a 6-layer GCN as described in B.1 with a 100 dimensional channel space and latent space. We use 1k training geometries and train the model for 20k iterations using the Adam optimizer (Kingma & Ba, 2014) with a learning rate of $10^{-3}$. We use a training batch size of 100.

**Pred:** Once trained, at inference time, in GABI-ABC we decode 10k samples in 100 batches of 100 samples. We keep 100 samples as being drawn from the posterior.

### B.3.2 Additional Results

In Figure 6 we show the results of inference with GABI-ABC on for the full field reconstructing from the small number of sparse noisy observations in red. In Figure 7 we show the two terms in the loss for training the GABI autoencoder as well as a histogram of the encoded data batch overlaid with the 1D standard normal distribution. We then test the performance of ABC sampling vs NUTS sampling as we vary the dimension of the latent space for the heat inversion task. The MAE, quality of UQ, and inference times are shown in Figure 8. In Figure 9 we perform an ablation study on the size of the channel space of the graph autoencoder for a fixed latent dimension of 100 for Bayesian inversion with both ABC and MCMC sampling. From this, we can assess the importance of network size for accuracy and efficiency.

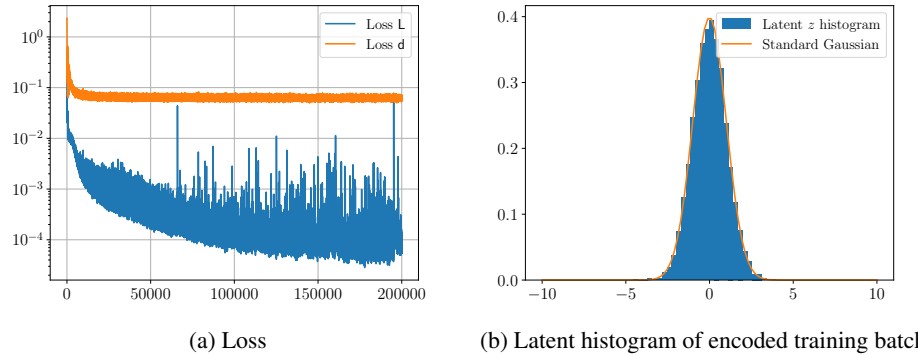

(a) Loss

(b) Latent histogram of encoded training batch

Figure 7: (a) Loss functions during training – the total loss is the sum of these. (b) histogram of the latent $p_{\mathbf{z}}^{\theta}$ across all dimensions after training.

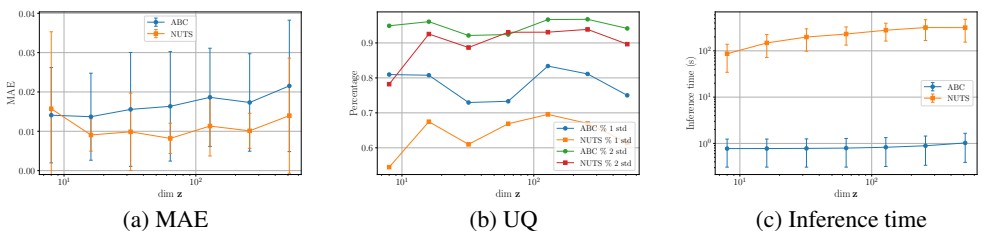

(a) MAE

(b) UQ

(c) Inference time

Figure 8: Sampling schemes on the same autoencoder model with the same channel space and different latent dimensions. We compare (a) mean absolute predictive error, (b) uncertainty quantification under a Gaussian assumption, (c) sampling time.

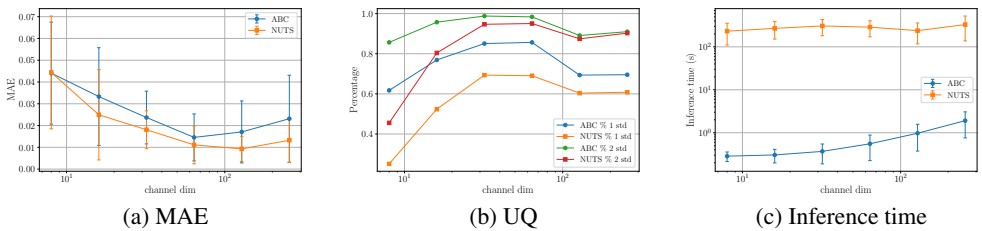

(a) MAE

(b) UQ

(c) Inference time

Figure 9: Sampling schemes on the same autoencoder model with different channel space dimensions and same the latent dimension (fixed at 100). We compare (a) mean absolute predictive error, (b) uncertainty quantification under a Gaussian assumption, (c) sampling time.

### B.3.3 TRANSFORMER GABI

As our framework is architecture-agnostic, we also test a Transformer-based variant. Transformers can function as neural operators (Cao, 2021), achieving non-locality through attention mechanisms. Unlike the graph-based approach, this variant treats the input as an unordered point set with coordinates, without exploiting the underlying graph structure. We implement 4-layer Transformers for the encoding and decoding networks with 64-dimensional embeddings and 8 attention heads per layer, resulting in a comparable number of trainable parameters: $\sim 200$k for GABI and $\sim 100$k for the direct map. As shown in Table 5, we obtain similar results to the GCN-based approach (with the same training setup), but much slower training and prediction times, which can be attributed to the dense self-attention operations of the Transformer layers.

Table 5: Comparison of Graph Convolutional Network (GCN) and Transformer (T) architectures: Heat Equation in Rectangular Domain

| Method | Field | MAE | % 1 std | % 2 std | Train | Pred. |
|--------|-------|-----|---------|---------|-------|-------|
| T: GABI-ABC | $(u)$ | $1.19 \cdot 10^{-2} \pm 8.65 \cdot 10^{-3}$ | 82.97% | 97.97% | 18.89hr | 10.17s |
| T: Direct Map | $(u)$ | $1.01 \cdot 10^{-2} \pm 6.69 \cdot 10^{-3}$ | – | – | 11.00hr | $0.0063s$ |
| GCN: GABI-ABC | $(u)$ | $1.58 \cdot 10^{-2} \pm 1.36 \cdot 10^{-2}$ | 80.91% | 95.59% | 2.62hr | 0.908s |
| GCN: Direct Map | $(u)$ | $1.25 \cdot 10^{-2} \pm 1.02 \cdot 10^{-2}$ | – | – | 1.47hr | $0.0029s$ |

## B.4 AIRFOIL

### B.4.1 IMPLEMENTATION DETAILS

**Train:** We use an 8-layer GCN as described in B.1 with a 100 dimensional channel space and latent space. We use 1k training geometries and train the model for 10k iterations using the Adam optimizer (Kingma & Ba, 2014) with a learning rate of $10^{-3}$ with an exponential decay learning rate with decay rate 0.999. We use a training batch size of 100.

**Pred:** Once trained, at inference time, in GABI-ABC we decode 50k samples in 100 batches of 500 samples. We keep 100 samples as being drawn from the posterior.

### B.4.2 ADDITIONAL RESULTS

In Figure 10 we show four samples from the full field dataset. In Figure 11 we show a complete example of results from the airfoil inversion task. We only measure the noisy pressure along the surface of the airfoil on the red scatter points and infer from this the full pressure field and 2D velocity field. In Figure 12 we show four random samples drawn from the generative prior model for a given geometry, demonstrating the physical coherence and diversity of the prior.

We now test the behaviour of GABI-ABC inference on 10 out-of-distribution airfoil geometries. In Figure 13 we plot a histogram of the density of the training dataset geometries and a selection of out of distribution airfoils. The x-axis shows the Mahalanobis distance (empirical covariance weighted distance of a point to the mean) between these shapes and the training dataset. The distance in geometry is computed using 9 summary statistics regarding airfoil thickness, camber, trailing-edge thickness, etc. We report the MAE and percentage of the ground truth between one and two standard deviations of the predictive posterior for pressure (reconstructed with GABI using 20 observations). These tests suggest that the GABI framework has some mild, but limited, robustness to prediction for outliers. Eventually, the miss-specification of the prior for such out-of-distribution geometries may lead to unreliable inference.

## B.5 CAR

For the car resonance and source localization problem we create the dataset by solving the damped Helmholtz equation (Liu et al., 2024) on a closed manifold

$$(\Delta_{\mathcal{M}_n} - \kappa + i\gamma\kappa) u(x) = f(x), \quad x \in \mathcal{M}_n. \tag{10}$$

where $i = \sqrt{-1}$, $\kappa = 500$ is a wave speed parameter, and $\gamma = 0.2$ is a damping parameter.

### B.5.1 IMPLEMENTATION DETAILS

**Train:** We use an 6-layer GCN as described in B.1 with a 100 dimensional channel space and latent space. We use 500 training geometries and train the model for 10k iterations using the AdamW optimizer (Loshchilov & Hutter, 2017) with an initial learning rate of $10^{-3}$ with and cosine decay learning minimum rate of $10^{-5}$. We use a training batch size of 100.

**Pred:** Once trained, at inference time, in GABI-ABC we decode 50k samples in 100 batches of 500 samples. We keep 100 samples as being drawn from the posterior.

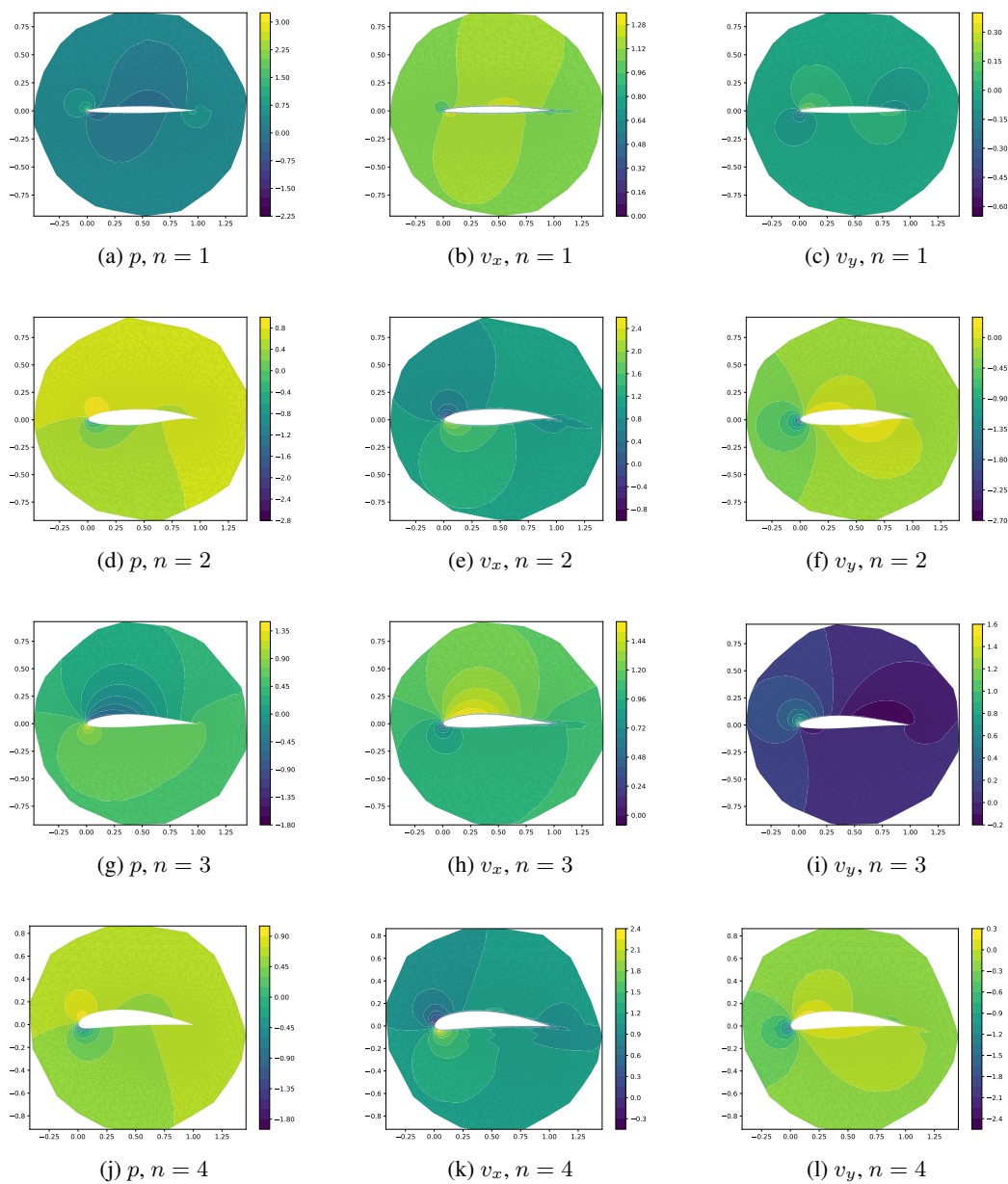

Figure 10: Four samples from the Airfoil dataset for pressure, horizontal and vertical velocity fields.

### B.5.2 ADDITIONAL RESULTS

In Figure 14 we show some example geometries along with the forcing function and resonance field to reconstruct. In Figure 15 we show 3 random draws from the prior for a given geometry to show the diversity of solutions and forcing fields.

## B.6 TERRAIN

### B.6.1 IMPLEMENTATION DETAILS

**Train:** For both the encoding and decoding layers of the autoencoder we use an encode-process-decode backbone (Pfaff et al., 2020; Duthé et al., 2025), where each processing block is composed

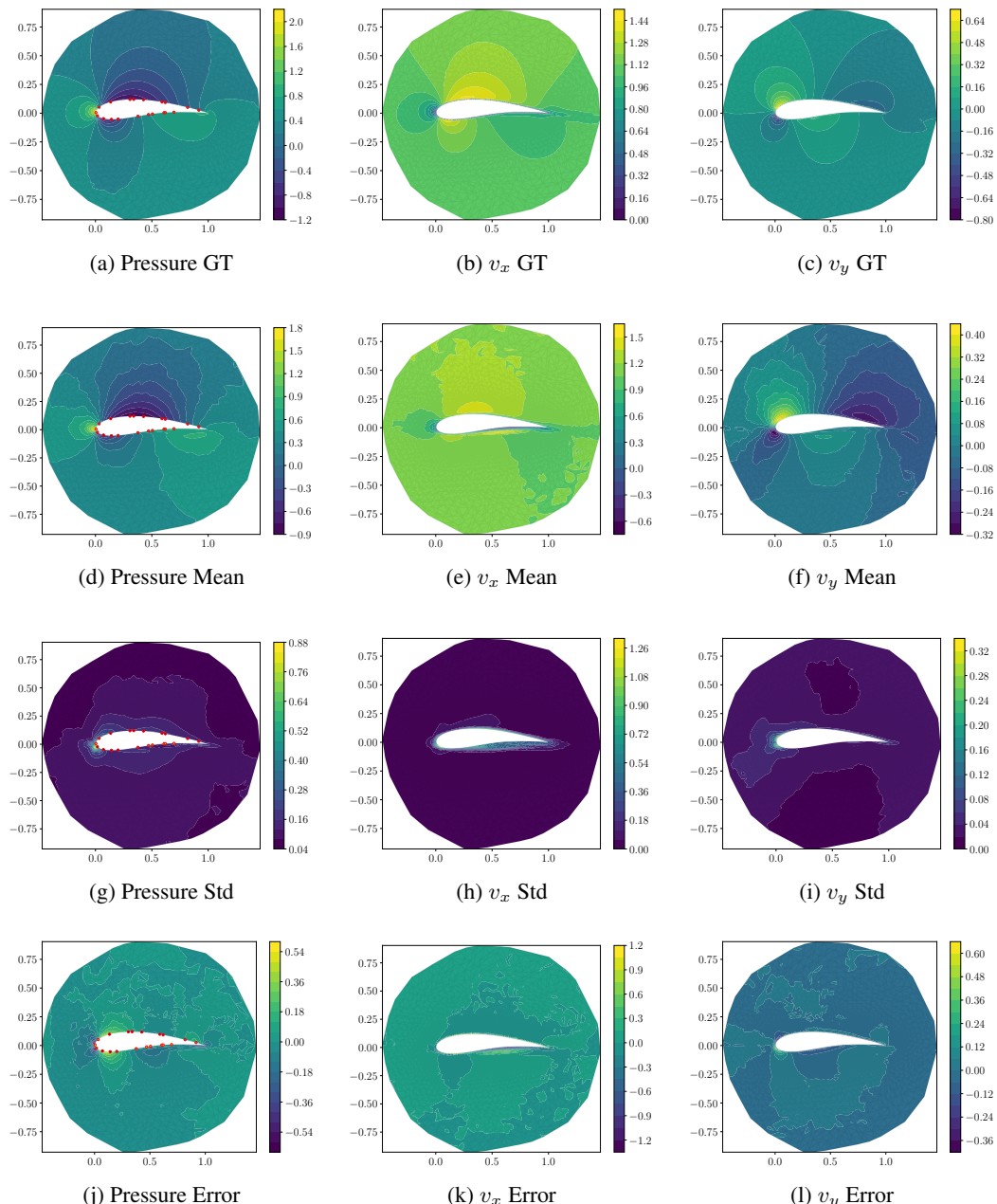

Figure 11: Comparison of ground truth (GT), inferred mean, error, and standard deviation for pressure ($p$), horizontal velocity ($v_x$), and vertical velocity ($v_y$). Columns group physical variables; rows show different prediction outputs.

of a 6-layer GEN-based GNN. Both edge and node features are projected to latent vectors with 64 dimensions. We train the model for around 300 epochs using the Adam optimizer with an exponential learning rate decay (initial learning rate of $10^{-3}$, decay exponent of 0.99). This takes around 52 hours on our distributed training pipeline across four RTX4090 GPUs with an overall batch size of 20 graphs.

We process the training samples from Achermann et al. (2024), which are larger ($96 \times 96 \times 64$) than the testing samples ($64 \times 64 \times 64$). Following their procedure, we apply random on-the-fly crops and rotations during training. We then further reduce the samples by keeping only the 8 fluid cell

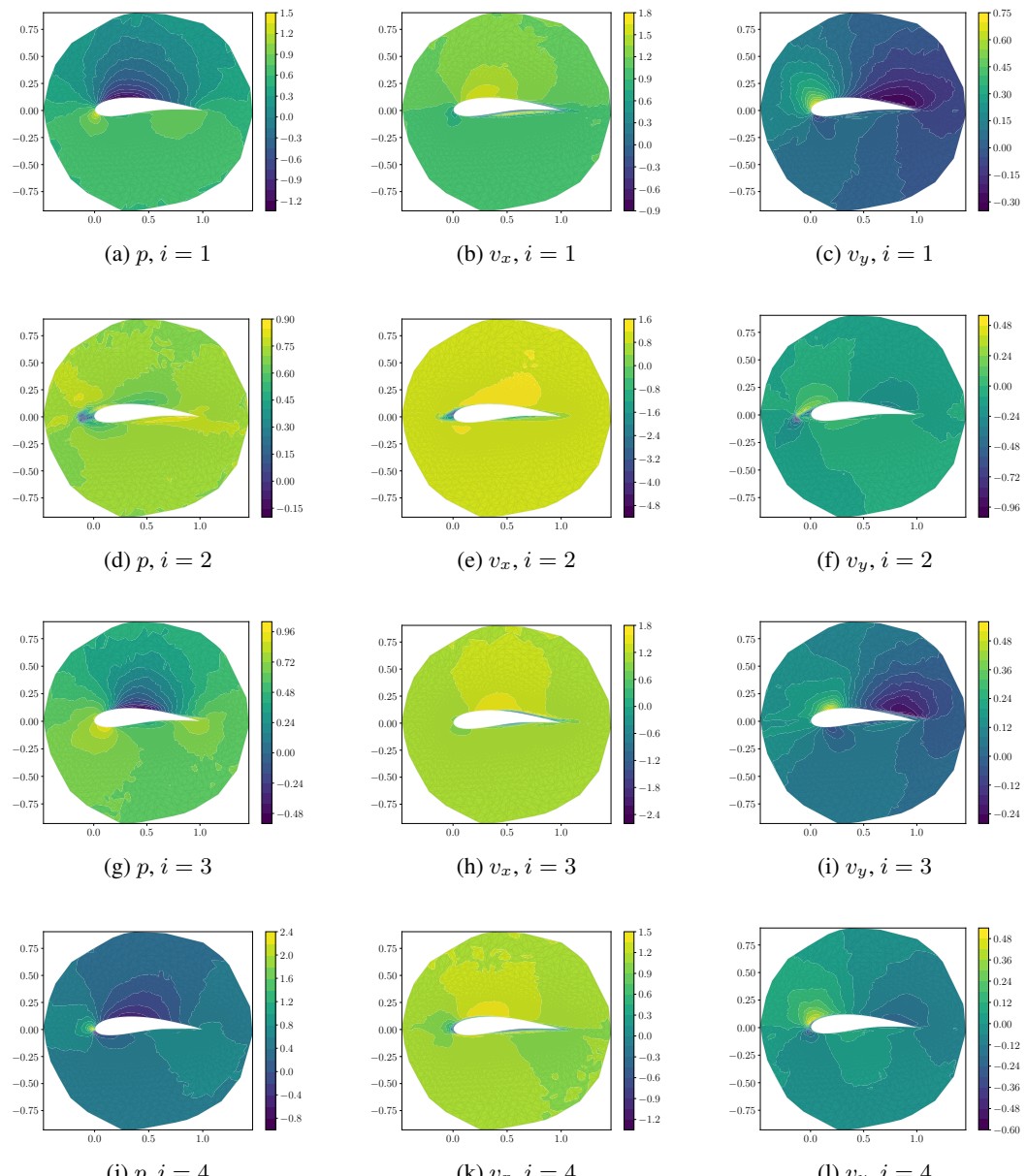

Figure 12: Airfoil field samples drawn from the geometry conditioned joint prior over pressure, $v_x$, and $v_y$ generated as $\mathbf{u}_i \sim D_{n \#}^{\psi} q_{\mathbf{z}}$

layers directly above the terrain, resulting in a final topography-conforming training geometry of $64 \times 64 \times 8$.

**Pred:** At inference time, we decode 50k samples in batches of 40 samples, distributed over 4 GPUs, which takes around 480 seconds. We keep 100 samples as being drawn from the posterior.

### B.6.2 ADDITIONAL RESULTS

In Figure 16 we show some examples of flows over different terrain geometries taken from the training set. In Figure 17 we show four random samples drawn from the generative prior model for a given terrain geometry, demonstrating the physical coherence and diversity of the prior. Due to the

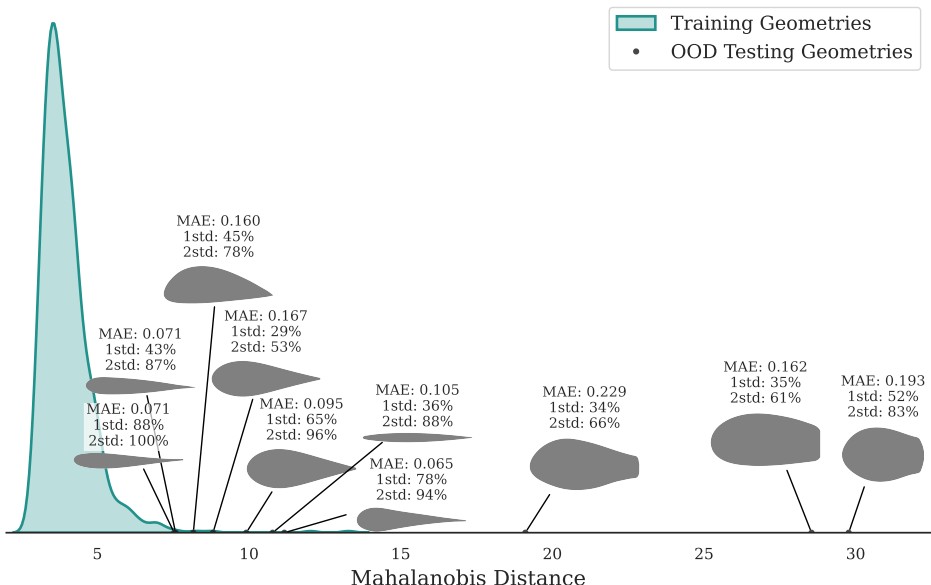

Figure 13: Out of distribution airfoil shapes and corresponding GABI inversion MAE and UQ for pressure, based on 20 observed airfoil nodes.

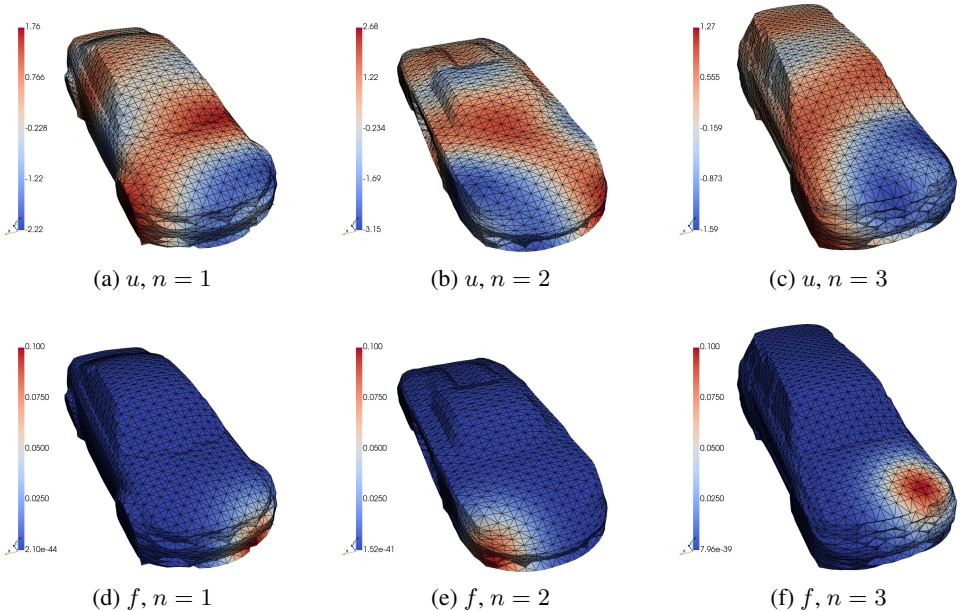

(a) $u, n = 1$      (b) $u, n = 2$      (c) $u, n = 3$

(d) $f, n = 1$      (e) $f, n = 2$      (f) $f, n = 3$

Figure 14: Three example meshes in dataset

nature of the training data (inflows from all directions are possible), the generated samples are more diverse in nature than the previous numerical setups.

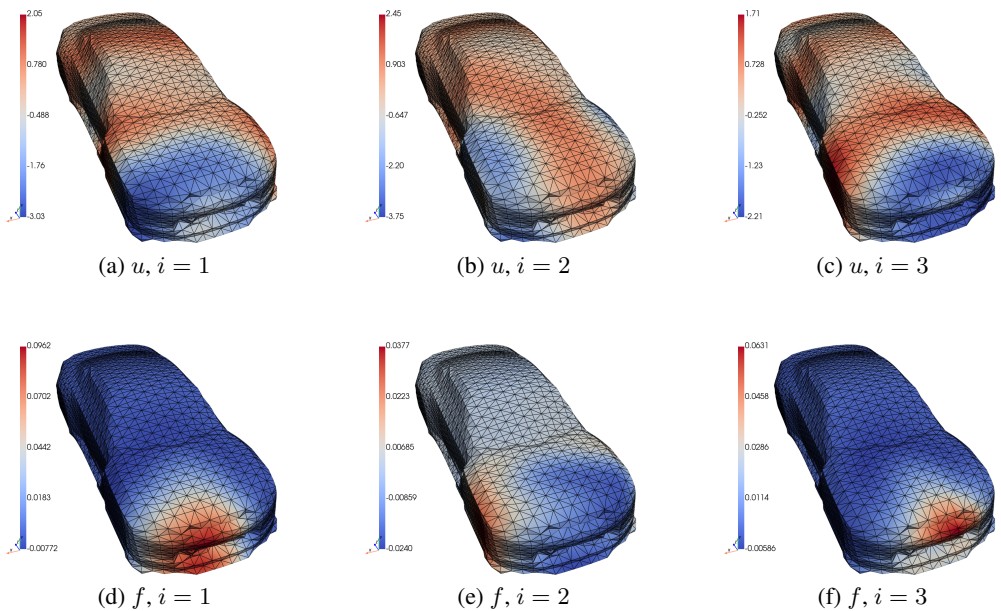

Figure 15: Given a geometry, we show 3 samples from the conditional prior generated as $\mathbf{u}_i \sim D_n^{\psi} \# q_{\mathbf{z}}$

## C    DISCUSSION ON VAES

### C.1    ALTERNATIVE VARIATIONAL FORMULATIONS

In this work we propose to: first train an autoencoder to learn an informative prior in the latent space, second to sample from the posterior given observational data. At first glance, an alternative approach would be to forgo this two step approach and attempt to directly learn a probabilistic geometry-conditional autoencoder in the form of a VAE

$$\theta^{\star}, \psi^{\star} = \arg\min_{\theta, \psi} \mathbb{E}_{\mathcal{D}_{\mathbf{y}}} \, \mathsf{KL}\left(q_{\mathbf{z}|\mathbf{y}_n}^{\theta} \big\| p_{\mathbf{z}|\mathbf{y}_n}^{\psi}\right), \tag{11}$$

$$\mathsf{KL}\left(q_{\mathbf{z}|\mathbf{y}_n}^{\theta} \big\| p_{\mathbf{z}|\mathbf{y}_n}^{\psi}\right) = \log p_{\mathbf{y}_n}(\mathbf{y}_n) + \mathbb{E}_{q_{\mathbf{z}|\mathbf{y}_n}^{\theta}}\left[-\log p_{\mathbf{y}_n|\mathbf{z}}^{\psi}(\mathbf{y}_n)\right] + \mathsf{KL}\left(q_{\mathbf{z}|\mathbf{y}_n}^{\theta} \big\| q_{\mathbf{z}}\right). \tag{12}$$

Upon close inspection of the loss for such a model, it is apparent that information from the full solution field $\mathbf{u}_n$ is not incorporated in the learning, hence one cannot hope to learn the correct relationship between $\mathbf{y}_n$ and $\mathbf{u}_n$.

One could alternatively train a model of the form

$$\theta^{\star}, \psi^{\star} = \arg\min_{\theta, \psi} \mathbb{E}_{\mathcal{D}_{\mathbf{y}}} \, \mathsf{KL}\left(q_{\mathbf{u}_n|\mathbf{y}_n}^{\theta} \big\| p_{\mathbf{u}_n|\mathbf{y}_n}^{\psi}\right). \tag{13}$$

However, as $\mathbf{u}_n$ are each associated with different geometries $\mathcal{M}_n$ we cannot put a common prior distribution over these in a statistically interpretable manner. Furthermore, such a model would not learn the correct relationship between $\mathbf{y}_n$ and $\mathbf{u}_n$ as observational data is ingested in an unsupervised manner; no regression loss encourages $\mathbf{y}_n$ to be close to $\mathbf{u}_n$ in any way.

There exist other models such as UQ-VAEs (Goh et al., 2022) where the decoder is replaced by the solution operator for the assumed PDE responsible for generating the data. This of course requires us to know the governing equations of the problem, the boundary conditions etc – and has the same challenges and limitations as other physics-informed methodologies. Furthermore, this framework is not made to handle variable geometries.

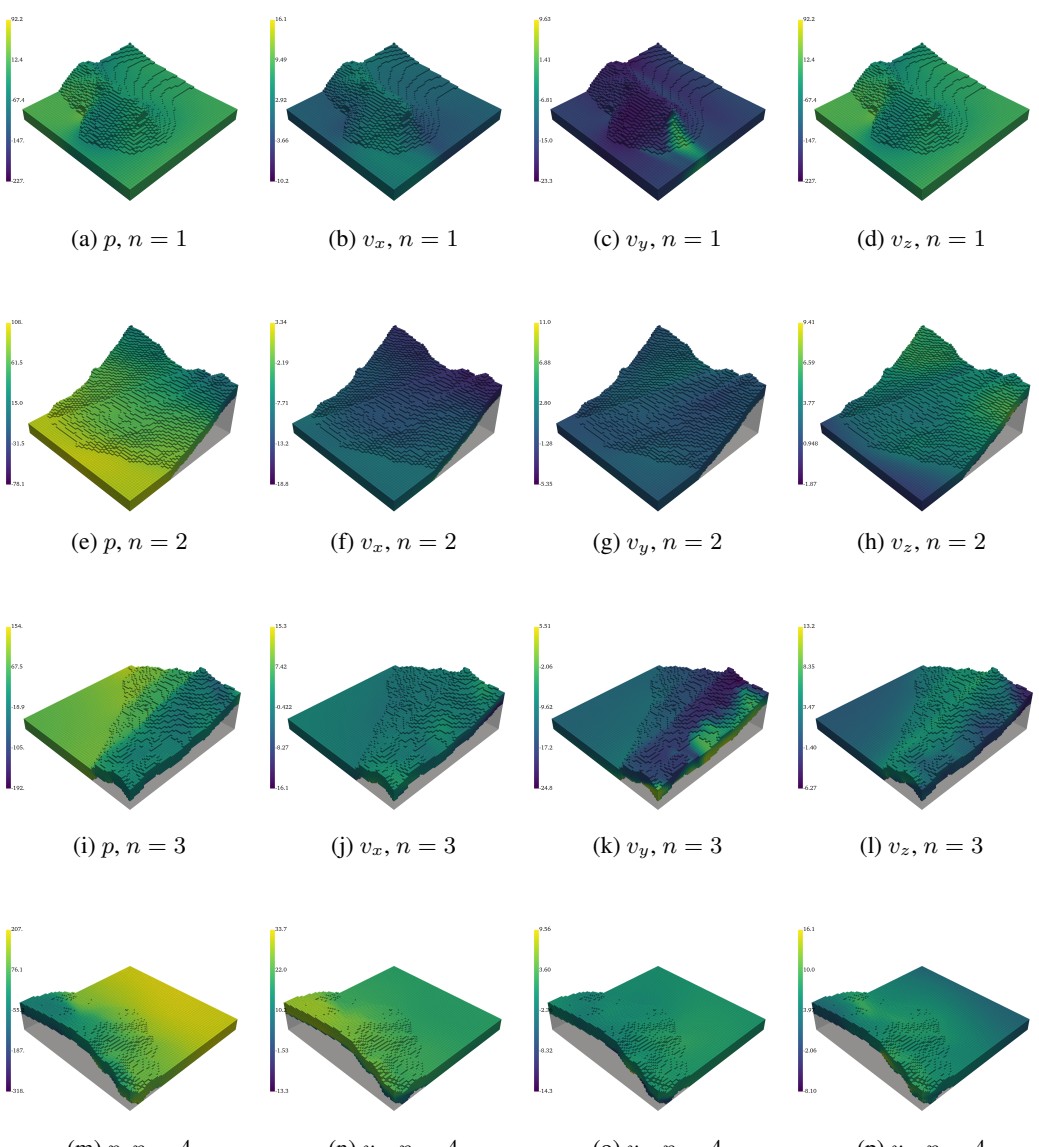

Figure 16: Four examples of flows ($p$, $v_x$, $v_y$, $v_z$) over complex terrain taken from the training set.

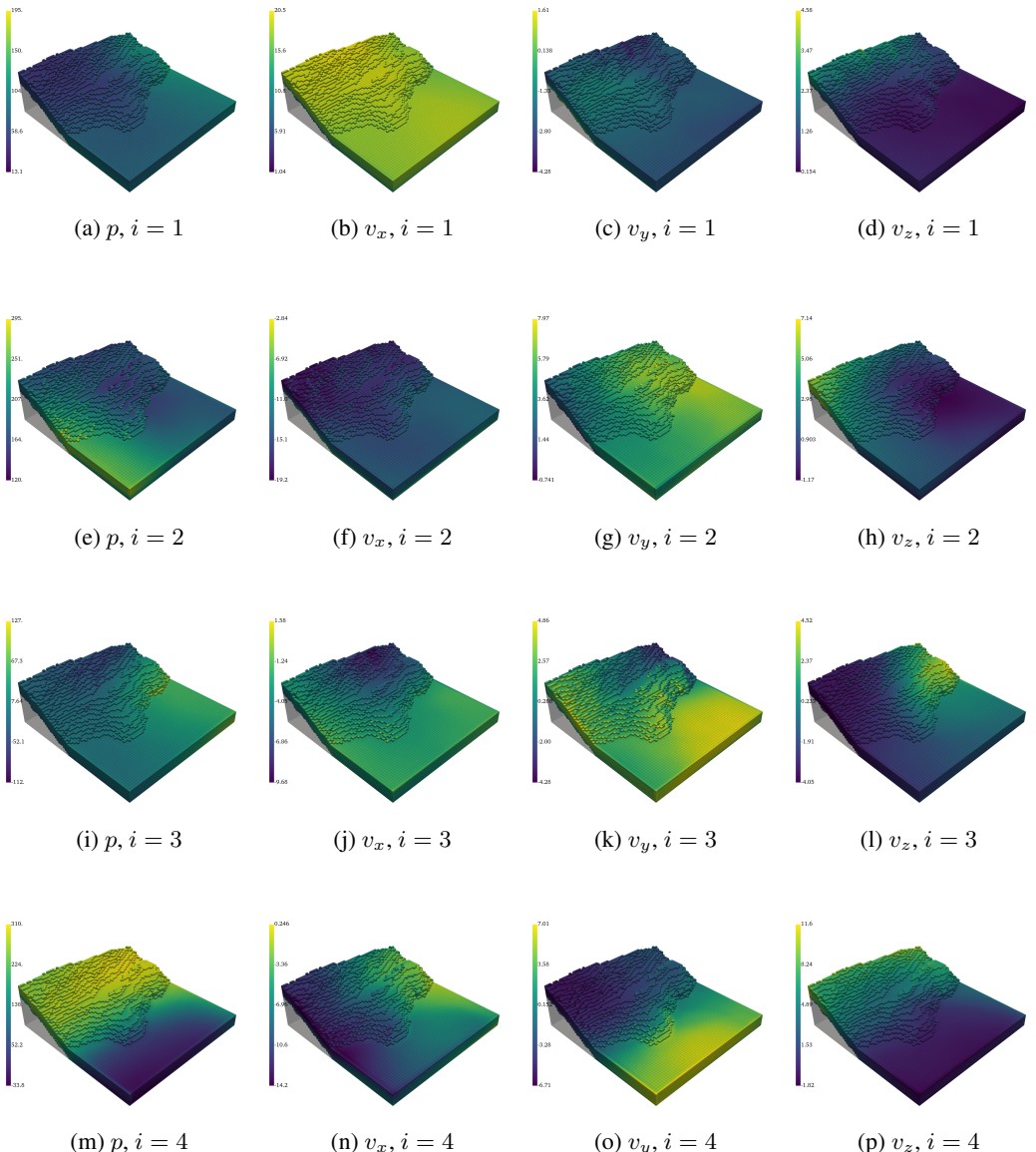

(a) $p$, $i = 1$   (b) $v_x$, $i = 1$   (c) $v_y$, $i = 1$   (d) $v_z$, $i = 1$

(e) $p$, $i = 2$   (f) $v_x$, $i = 2$   (g) $v_y$, $i = 2$   (h) $v_z$, $i = 2$

(i) $p$, $i = 3$   (j) $v_x$, $i = 3$   (k) $v_y$, $i = 3$   (l) $v_z$, $i = 3$

(m) $p$, $i = 4$   (n) $v_x$, $i = 4$   (o) $v_y$, $i = 4$   (p) $v_z$, $i = 4$

Figure 17: Flow field samples for a given test terrain drawn from the geometry conditioned joint prior over $p$, $v_x$, $v_y$, and $v_z$, generated as $\mathbf{u}_i \sim D_{n\,\#}^{\psi} q_{\mathbf{z}}$

