# OpenReview forum: "Geometric Autoencoder Priors for Bayesian Inversion: Learn First Observe Later"
_ICLR.cc/2026/Conference — ICLR 2026 Poster_

### Official Review · Reviewer_TrCV · 2025-10-15

**Soundness:** 1
**Presentation:** 1
**Contribution:** 3
**Rating:** 2
**Confidence:** 2

**Summary:**

The paper proposed an autoencoder-based scheme of constructing priors for inverse Bayesian problems in PDE. These priors are build from data of similar tasks with the same underlying physics but different meshes.

**Strengths:**

The strength in this paper lie in strong experimental evaluation. The results of the experiments seem good and improve upon the state of the art by up to an order of magnitude. The Experimental range is quite wide, starting from the simple heat equation to more complex, non-linear systems of PDEs in varying geometries. This is the main reason for a good contribution score.

**Weaknesses:**

The main weakness lies int he presentation of the paper. I had quite a hard time of following it and even at this stage I do not get the details of the approach.
 - Some problems lie in formalizing the notation. See the questions below. I stress that these questions are just for the main paper, the appendix (which I just flew over) was equally problematic. I want to clarify that for many of my questions bwloe it is alluded what the notation means, but not made formal.
 - The paper solves a relevant problem. However, it took me quite some time to understand (and then only even vaguely) which problem is being solved. My main problem was that it took some time to understand what was given and what was not given.
 - Experiments compare to machine learning methods only, and not to methods from the inverse problems literature. (I know that it is hard to do experiments against all potential competitors, so I am not dwelling on this point for so long, as some comparisons are being made.)
 - I did not find any description of computational time in this paper. For such computational approaches, I think this is a critical point.
 - l. 256ff "The proposed methodology is agnostic to any particular architecture, however we need the following characteristics: Geometry aware architecture and data-structure; Mapping to and from fixed-dimensional vectors; Non-locality. We expand on these points in B." This makes the paper (without appendix) far from being standalone.
 - Lemma 2.1 is relatively simple. However, it is stated very technically, without giving an intuition.
These weaknesses explain my soundness and presentation scores.

**Questions:**

l. 96: which space is $u_o$ from?
l. 97: What kind of Operator is $H_o$? Linear? Bilinear?
l. 97: $R^o$ is the output of $H_o$. Why/how does it select nodes?
l. 198: How do you compute the argmin? (SGD does not compute argmin!)
l. 207: what is $H_o$? What is $D_o$?
l. 211-215: what are the various $p$'s introduced here?
l. 255ff: I do not even know which part of the methodology is trained using neural networks. Encoder? Decoder? $H_o$? All of them?

---

> ### Author Response · Authors · 2025-11-19
> **Response: Clarity, Typos, Exposition, Part I**
>
> We thank the reviewer for their comments and feedback about our manuscript. We will answer each comment individually and reference where these comments and questions have inspired changes to our manuscript in the uploaded revision. We have decided to use the discussion/rebuttal phase extra page to clarify communication, expand on explanations (particularly regarding technical sections) and overall improve the accessibility of the paper.
>
> As the reviewer agrees the experiments are strong, varied, demonstrate good performance; and in light of the other reviewers unanimously agreeing on the relevance and value of this work; we kindly ask if the reviewer would consider having a look at our updated manuscript and consider revising their score.
>
> We believe this work directly addresses a significant and important research gap in ML and engineering and many members of the ICLR community, in particular in the engineering sciences, would benefit from reading our work.
>
> **Comments and Responses/Edits.**
>
> i) **However, it took me quite some time to understand (and then only even vaguely) which problem is being solved. My main problem was that it took some time to understand what was given and what was not given.**
>
> Thank you for this comment, we have added a bullet point description of the methodology and what is given at what stage. These are at the top of page 2. The addition reads:
>
>  *Hence, the proposed methodology works as follows:*
> - *In the training stage, using a dataset of geometries and full-field solutions on these geometries, $\mathcal{D}$ = {$ \mathbf{u}_n, \mathsf{M}_n$}$\_{n=1}^N$, we learn a graph autoencoder which embeds the joint distribution of solutions and geometries into a latent Gaussian distribution. This will act as our  highly informative geometry-aware data-driven prior.*
> - *In the application stage, we use this learned prior to solve inverse problems for data coming from new physical systems not in the dataset $\mathcal{D} $. That is, given sparse and noisy observations $\mathbf{y}_o$ of a solution $\mathbf{u}_o$ of a physical systems with geometry $\mathsf{M}_o$, we use our previously learned prior to solve the Bayesian inverse problem for the unknown full-field $\mathbf{u}_o$. We recover a distribution over the solution $\mathbf{u}_o$ by only observing $\mathbf{y}_o$.*
>
> ii) **Experiments compare to machine learning methods only, and not to methods from the inverse problems literature. (I know that it is hard to do experiments against all potential competitors, so I am not dwelling on this point for so long, as some comparisons are being made.)**
>
> We appreciate this comment. For an inverse problem over full-fields, a natural choice of prior is a Gaussian process prior. As the observation process $\mathbf{H}_o$ is linear, the classical Bayesian posterior for this problem is precisely a Gaussian process regression. Hence, our comparison to GPs is, in some sense, a comparison to a classical inverse problem method. We perform hyper-parameter optimization of this GP, which relates it also to modern uses of GPs in ML, but as these seem to improve their accuracy in this context, we deem this to be a more fair comparison.
> We agree we have not explicitly mentioned this link in the paper. From this discussion, we have added this discussion in Section 3, last paragraph.
>
>
> iii) **I did not find any description of computational time in this paper. For such computational approaches, I think this is a critical point.**
>
> Thank you for pointing this out. These are wall-clock times. We have added a short description of the reported training time:
>
> *In all experiments we report the wall-clock training time (Train) on Nvidia RTX 4090 GPUs (only the terrain problem is multi-GPU), and the mean per-geometry prediction time (Pred.).*
>
> iv) **l. 256ff "The proposed methodology is agnostic to any particular architecture, however we need the following characteristics: Geometry aware architecture and data-structure; Mapping to and from fixed-dimensional vectors; Non-locality. We expand on these points in B." This makes the paper (without appendix) far from being standalone.**
>
> As is standard in venues such as ICLR, the appendix is essential in providing readers with all available information for maximizing reproducibility. We have chosen to defer this discussion to the appendix as it focuses on extending our work and does not serve to convey the results we have obtained. Similarly we defer detailed descriptions of architectures to the appendix.

---

> > ### Author Response · Authors · 2025-11-19
> > **Part II**
> >
> > v) **Lemma 2.1 is relatively simple. However, it is stated very technically, without giving an intuition. These weaknesses explain my soundness and presentation scores.**
> >
> > Thank you for this comment. Using the allotted extra page we have added a remark directly below Lemma 2.1 to help guide readers who might not be familiar with measure-theoretic notation. We would like to point out, this notation is necessary for this central result, as proofs based on probability densities require extra conditions on $g$ which cannot be expected to hold.
> > The added remark reads:
> > *Stated simply, in Bayesian inference, when a prior on $\mathcal{U}$ is specified as the pushforward under $g$ (Subsection 1.3) of a latent prior on a space $\mathcal{Z}$, two distributions are equivalent: i) the Bayesian posterior on $\mathcal{U}$, and ii) the pushforward under $g$ of the posterior associated to the latent prior on $\mathcal{Z}$. This result will be explicitly related to the geometric inference setting in Theorem 2.2, showing that with a latent prior constructed via a geometry aware encoder/decoder, we can solve inverse problems on $\mathcal{U}$ with a single common latent representation relevant to any number of in-distribution geometries.*
> >
> > vi) **l. 96: which space is $\mathbf{u}_o$ from?**
> >
> > vii) **l. 97: What kind of Operator is H_o ? Linear? Bilinear?**
> >
> > viii)**l. 97: R_o is the output of H_o. Why/how does it select nodes?**
> >
> > Thank you for pointing this point, we have made a typo. We mean to write: $\mathbf{H}_o:\mathbb{R}^{d_u}\rightarrow\mathbb{R}^{o}$. (So $\mathbf{H}_o$ being linear is in $\mathbb{R}^{d_u\times d_o}$). Implicit in this correction is $\mathbf{u}_o\in\mathbb{R}^{d_o}$. Furthermore, to clarify point viii) we have added on page 1:
> >
> > *In this work, we take $\mathbf{H}_o:\mathbb{R}^{d_u}\rightarrow\mathbb{R}^{d_o}$ to be a diagonal matrix of zeros and ones selecting the nodes to be observed. This is assumed to be a known observation operator corresponding to sensor placements. We note that many observation models can be considered with ease, such as surface integrals for computing drag coefficients, radon transforms for tomography, as well as any number of noise models -- additive or not. As our learned prior is entirely independent of the observation process, changes to the observation process do not necessitate retraining and can be incorporated ``on-the-fly''.*
> >
> >
> > ix) **l. 198: How do you compute the argmin? (SGD does not compute argmin!)**
> >
> > Indeed we minimize this objective with gradient descent as specified in the numerics section. We add below (3):
> >
> > *argmin will be approximated with gradient descent.*
> >
> > If the reviewer has a different view of argmin and it's relation to SGD, please let us know. We are happy to make further changes.
> >
> > x) **l. 207: what is H_o? What is D_o?**
> >
> > $\mathbf{H}_o$ is the observation operator associated to the observation vector $\mathbf{y}_o$.
> >
> > $D_o$ is the decoder acting on a latent vector $\mathbf{z}$ conditioned on a graph $\mathsf{M}_o$. We have moved the statement  "where $D^\psi_o = D^\psi(\cdot;\mathsf{M}_o)$" from theorem 2.3 to below equation (4) so the reader can see this more easily.
> >
> > The reason for using indexing under $E$ and $D$ (with $o$ for graphs not in the dataset and $n$ for graphs in the dataset) is to lighten notation as not have explicit references to $\mathsf{M}_{n/o}$ whenever we decode or encode.
> >
> > xi) **l. 211-215: what are the various p's introduced here?**
> >
> > We have updated the preceding sentence to explicitly reference these distributions:
> >
> > *We specify our prior distribution over $\mathbf{u}\_o$, denoted $p\_{\mathbf{u}\_o}$, using the trained decoder; and specify the likelihood, denoted $p^{\psi}\_{\mathbf{y}\_o|\mathbf{z}}$, associated to the decoded observation model (4).*
> >
> >
> > xii) **l. 255ff: I do not even know which part of the methodology is trained using neural networks. Encoder? Decoder? ? All of them?**
> >
> > $E^\theta$ and $D^\psi$ are encoder decoders respectively and are implemented through graph neural networks.
> >
> > We clarify this point by adding the sentence (Section 2.2 Step 1):
> >
> > *Both $E^\theta$ (the encoder) and $D^\psi$ (the decoder) are graph neural networks with parameter sets $\theta, \psi$.*

---

> > > ### Comment · Reviewer_TrCV · 2025-11-22
> > >
> > > I had a look at the updated paper. Several more formal problems were obvious.
> > >
> > >  - The distinction between continuous function spaces and their discrete mesh representations is blurred. For example, $u_n \in U_n := U(M_n;\mathbb{R}^{d_u})$ is introduced as a discretized object, although $U(M_n;\mathbb{R}^{d_u})$ is a function space.
> > >
> > >  - The notation $d_n$ (for the number of vertices?) is used without formal definition.
> > >
> > >  - The posterior $p_{z\mid y_o}^\psi$ is defined with a superscript $\psi$, but later expressions use $p_{z\mid y_o}$ without it.
> > >
> > >  - The paper mixes measures $(P_u,P_z)$ and densities $(p_u,p_z)$ in a way that is sometimes ambiguous. (I might be nitpicky here.)
> > >
> > >  - In the noise-estimation section, the joint posterior $p_{z,\sigma\mid y_o}(z,\sigma)$ is written with a prior $p_{z,\sigma_o}(z_o,\sigma)$, introducing new subscripts ($z_o,\sigma_o$) without definition, and the notations $\sigma$ and $\sigma_o$ are used inconsistently.
> > >
> > >  - In the experimental sections there are inconsistencies in the use of indices $n$ and $o$ for training and test geometries, and some key constructs are only loosely specified.
> > >
> > >  - In the GP baselines it is not made precise whether the kernel is defined on node coordinates, graph structure, or something else.
> > >
> > > I do not intend to read the paper again, as I would guess to find more formal problems in another readthrough. I leave it to the AC to decide whether to trust the presentation of the paper. I will keep my score.

---

> > > > ### Author Response · Authors · 2025-11-24
> > > > **Follow-up Response**
> > > >
> > > > We thank the reviewer for their comments. We have made some changes to our manuscript:
> > > >
> > > > - We have corrected the subscript inconsistencies, and added the $\psi$ superscript where missing.
> > > > - We explicitly introduce the notation in the noise estimation section.
> > > > - We reiterated in the numerics section that the GPs are defined over graphs, (this was mentioned in the literature review, which we agree ought to be specified in the numerics section as well).
> > > >
> > > > As for the other points, these are choices we have made to lighten the paper, ex: we only distinguish between function spaces and discretized spaces with fonts, as these concepts are intimately related in the context of this paper. Also, as mentioned in the notation section 1.3, when using lower case $p$ we refer to the density of the measures $\mathbb{P}$ with the same label.
> > > >
> > > > We would appreciate any other comments or suggestions the reviewer might have.
> > > >
> > > > For the AC’s clarity, we note that all points raised in the reviewer’s follow-up message concern notation consistency and presentation.
> > > > These edits do not affect the soundness of any theorem, algorithm, experiment, or claim.

---

### Official Review · Reviewer_RTKB · 2025-11-01

**Soundness:** 3
**Presentation:** 3
**Contribution:** 3
**Rating:** 6
**Confidence:** 4

**Summary:**

This paper combine autoencoder (AE) and approximate Bayesian computation (ABC) for inverse problems. AE is trained on geometry of the physics problems to obtain common features in the latent space, which is independent of the governing equations (PDE). The observation process is only used in the sampling stage with ABC. The proposed method, hence named Geometric Autoencoders for Bayesian Inversion (GABI), enables efficient Bayesian inversion in a `learn first, observe later` fashion.

**Strengths:**

The idea is straightforward, clearly presented, and efficiently implemented (via multi-GPUs). The demonstration on high-dimensional real engineering problems (car body acoustic vibration) is definitely a plus.

**Weaknesses:**

It is not clear the trade-off between the size of geometric AE and the precision of the downstream ABC. And it is not clear how it compares with transfer learning method.

**Questions:**

1. How large the geometric AE is required to ensure sufficient precision in ABC? Is there any trade-off between the size of AE and accepted sample size $N_a$? Can you do some ablation study on it?

2. Since the AE is trained on the geometry of different but similar physics problems, I wonder how GABI relates to or compares with transfer learning methods?

Minor things:

Line 062: remove semicolon.
Line 098: I assume you meant for $H_o: R^{d_u} \to R^{d_o}$?
Line 119, 143: period is missing after the reference.
Line 422: `Figure` 4.
After Equation (6), maybe emphasize that the likelihood is not computed but ABC is used.

---

> ### Author Response · Authors · 2025-11-19
> **Response: Ablation, Transfer Learning, Comments**
>
> i) **How large the geometric AE is required to ensure sufficient precision in ABC? Is there any trade-off between the size of AE and accepted sample size? Can you do some ablation study on it?**
>
> Thank you for this question. Similar to our ablation study on the size of the latent space, with fixed network channel dimension, we have now run an ablation study on the effect of the size of the channel space of the network, keeping the latent dimension fixed. This new experiment is in section B.3.2. Text addition reads:
>
> *In Figure 9 we perform an ablation study on the size of the channel space of the graph autoencoder for a fixed latent dimension of 100 for Bayesian inversion with both ABC and MCMC sampling. From this, we can assess the importance of network size for accuracy and efficiency.*
>
> As a small point of clarification, the AE does not effect the accepted sample size. As shown in Algorithm 1, we specify the sample budget and the number of accepted samples --  this in turn implies an effective ABC tolerance. As the acceptance tolerance holds no intrinsic value, we only know that we want it to be as small as our resources allow, we find this to be a more convenient parametrization of ABC.
>
> The principle way we have of knowing if a network performs adequately for our inference task without having access to some ground truth, is by assessing the quality of reconstruction for a holdout set, as well as how close the embed hold-out dataset is to being Gaussian of the latent space which can easily be assessed qualitatively and quantitatively. This is in fact how we selected best architectures and latent dimension etc. We clarify this point in the manuscript in Section 4.
>
>
> ii) **Since the AE is trained on the geometry of different but similar physics problems, I wonder how GABI relates to or compares with transfer learning methods?**
>
> Thank you for this question. In our comparisons we consider two, rather opposite alternatives to GABI: zero-transfer Gaussian process regression, and (what is essentially pure transfer) Direct Regression. Overall, it seems much of the transfer learning literature is more of a conceptual paradigm rather then a specific strict mathematical model. In this sense, by learning a latent representation which extracts features from distinct but related geometries and PDE parametrizations, one could argue GABI is a form of Bayesian transfer learning. Related to this are Bayesian hierarchical models which share information from many systems, however these will of course not permit variations in geometry the way GABI does. If the reviewer has specific example of transfer learning in mind which might be applicable, we are happy to incorporate discussion of these into the paper.
>
>
> iii) **Line 098: I assume you meant for $H_o: \mathbb{R}^{d_u}\rightarrow\mathbb{R}^{d_y}$?**
>
> We did indeed, thank you. This has been updated, and we've added clarifying information following.
>
> iv) **After Equation (6), maybe emphasize that the likelihood is not computed but ABC is used.**
>
> Thank you for this suggestion. The method we propose, can be used with MCMC, which would indeed use the computations of the likelihood. In Table 1 and Figure 8 we compare the use of ABC with MCMC. We take note of this suggestion and highlight this in the first paragraph of Section 4:
>
> *We also note that in using ABC for sampling from the posterior, we do not need to compute the likelihood function. This may have strong advantages in application where the observation process yields intractable likelihoods. In using MCMC, as in the comparisons, the likelihood is computed.*
>
> Please find attached the updated manuscript. We believe this has improved the quality and clarity of our work. If these updates and changes answer the reviewers (and other reviewers) points well, we would kindly ask if they would consider updating their score to reflect this.

---

> > ### Comment · Reviewer_RTKB · 2025-11-25
> >
> > Thanks to authors for their elaboration. I appreciate the added ablation study.

---

### Official Review · Reviewer_WWiD · 2025-11-03

**Soundness:** 4
**Presentation:** 3
**Contribution:** 4
**Rating:** 6
**Confidence:** 4

**Summary:**

This is an interesting work applying Bayesian methods (autoencoders) to Geometric (prediction) problems. This will be applied when a new shape is designed, and people want to know how the field (temperature, pressure) will change. The authors use a simple solution that just encode the shapes (M) into a latent space during learning. When given a new shape, suppose we have enough trained shapes - the model will sample in the latent space N(0, 1), and produce related vector fields from the decoder, and match with the available observations. From four experimental study, the performance is verified.

**Strengths:**

I like this paper. It proposed a very valid method using autoencoders, and potentially solves a pretty difficult and highly practical problem.

1. The presentation is very nice. The written style, mathematical notation, experimental figures are all well presented.

2. The idea is clean, and nicely described. The algorithm is simple and easy to understand.

3. The experimental study nicely demonstrates the advantage of this method.

**Weaknesses:**

The theory part looks strong, but may have very limited theoretical impact. Although everything is well documented in theory, but it could create problems for readers. And an important critique from the reviewer is that - the theory seems to be rooted on two things (i) such latent space exists and (ii) the decoder (from this latent space) can be expressive enough, and it is able to nicely extrapolate data. These are all likely to be true, yet unfortunately, these assumptions are potentially hidden in the problem statement.

Please see questions below as well.

**Questions:**

1. In 1.1 setup, would the authors define U? In Lemma 2.1 or 1.3, would the authors use a line or two to describe what z normally is (e.g. the latent space)?

2. In Figure 2, the reviewer would like to ask (i) why is this figure so small.. and (ii) how should we interpret (b)?

3. The reviewer is very curious if the computation can be accelerated without incorporating the Bayesian framework. It seems that the Bayesian framework is important, but might not be that essential (if one wants to have the fastest algorithm).

4. How would GABI perform for OOD shapes? The reviewer believes that OOD it will perform poorly as the decoder can have random behavior during extrapolation. A comment here would be great (don't need to change paper).

5. Would the authors add some words about the implicit assumption in the main paper? As the latent space z might not actually exist. Or - the latent space z might be able to represent U/M under a bounded area/domain. With better/larger collected data, this problem can be resolved.

The reviewer is likely to improve the rating if some of the comments above being resolved.

---

> ### Author Response · Authors · 2025-11-19
> **Response: Clarifications, OOD Geometry, Interpretation of Figures, Part I**
>
> We thank the reviewer for their careful consideration of our manuscript as well as their comments and questions.
>
> **the theory seems to be rooted on two things (i) such latent space exists and (ii) the decoder (from this latent space) can be expressive enough, and it is able to nicely extrapolate data.**
>
> The  theoretical results presented, which underpin the method are indeed only related to idealized settings. They are expressions of the equivalence of two posterior on difference space related by some function which decodes one space to the other. Whether this program can be executed successfully in practice is not covered by this. Hence in practice one must first verify such a latent space can achieve this. To this end we have added explanations in Section 4 on how one might check whether one can expect GABI to work for a particular dataset and graph neural networks:
>
> *In GABI, the main way we have of knowing if a network performs adequately for our inference task without having access to some ground truth, is by assessing the quality of reconstruction for a holdout set, as well as how close the embed hold-out dataset is to being Gaussian of the latent space which can easily be assessed qualitatively and quantitatively.*
>
>
> i) **In 1.1 setup, would the authors define U? In Lemma 2.1 or 1.3, would the authors use a line or two to describe what z normally is (e.g. the latent space)?**
>
> We add in 1.1:
>
> *The space $\mathcal{U}$ is the relevant function space for the full field solution, e.g. a Sobolev space, and the geometric domain $\mathcal{M}_n$ may be a manifold with boundary or a boundaryless closed manifold.*
>
> As for the second point, we add a remark 2.2 below Lemma 2.1 which clarifies this:
>
> *Stated simply, in Bayesian inference, when a prior on $\mathcal{U}$ is specified as the pushforward under $g$ (Subsection 1.3) of a latent prior on a space $\mathcal{Z}$, two distributions are equivalent: i) the Bayesian posterior on $\mathcal{U}$, and ii) the pushforward under $g$ of the posterior associated to the latent prior on $\mathcal{Z}$. This result will be explicitly related to the geometric inference setting in Theorem 2.2, showing that with a latent prior constructed via a geometry aware encoder/decoder, we can solve inverse problems on $\mathcal{U}$ with a single common latent representation relevant to any number of in-distribution geometries.*
>
>
> ii) **In Figure 2, the reviewer would like to ask (i) why is this figure so small.. and (ii) how should we interpret (b)?**
>
> Indeed, the space constraint for the paper is a challenge. We use the extra page of the rebuttal phase to increase the size of 2 (a)-(b).
>
> *Figure 2 (b) shows 4 coloured histograms corresponding to the marginal distribution of the decoded posterior at the 4 queried location on Figure 2 (a). As the orange dot is close to a boundary with value 0 in the entire dataset (4 representative samples can be seen in Figure 1), GABI present very small uncertainty in this area. As we query points in areas where there is more variability of solution, GABI presents wider uncertainty. In all cases, the mean of the GABI posterior is very close to the ground truth (vertical dashed line).  This exemplifies the balance GABI strikes between sparse data and informative data-driven priors.*
>
> We add this discussion to Section 4.1.
>
> iii) **The reviewer is very curious if the computation can be accelerated without incorporating the Bayesian framework. It seems that the Bayesian framework is important, but might not be that essential (if one wants to have the fastest algorithm).**
>
>  One option would be to forgo sampling, and compute the maximum a posteriori estimate (MAP) of the solution field using optimization. Hence one would only recover a point solution. The optimization problem may be quite challenging as there may be many local minimums. A simulated annealing approach could possibly work and this would be compatible with the GABI framework, this would still incorporate the Bayesian framework in the sense of likelihood and prior, but would not compute UQ.

---

> > ### Author Response · Authors · 2025-11-19
> > **Part II**
> >
> > iv) **How would GABI perform for OOD shapes? The reviewer believes that OOD it will perform poorly as the decoder can have random behavior during extrapolation. A comment here would be great (don't need to change paper).**
> >
> >  Indeed we fully agree. As stated in the last paragraph of section 1.1, the idea behind GABI is to learn a highly informative prior from a dataset. The prior assumption is that the new geometry belongs to the data distribution. Hence, if a new geometry is presented which does not belong in some sense to the data distribution, GABI will not necessarily provide a relevant prior.
> >
> > How far one can stray from the data distribution is of course an interesting question which will depend lot on the architecture used and the effect of the changes on the physical dynamics (which can often be drastic!). We see the strength of GABI as it's ability to coherently synthesizing  relevant data to solve distinct but related problems. In an effort to explore GABI's out-of-distribution (OOD) capabilities, we have added an additional airfoil experiment in section B.4.2. We take 10 unseen OOD airfoil shapes (OOD based on key airfoil statistics) and reconstruct flows based on 20 observed nodes. We see mild but limited robustness to outliers. We also note that eventually, the miss-specification of the
> > prior for such OOD geometries may lead to unreliable inference.
> >
> > v) **Would the authors add some words about the implicit assumption in the main paper? As the latent space z might not actually exist. Or - the latent space z might be able to represent U/M under a bounded area/domain. With better/larger collected data, this problem can be resolved.**
> >
> > We hope our answer to the first comment has answered this question sufficiently. We are happy to expand on this if the reviewer has something else in mind.
> >
> > Please find attached the updated manuscript. We believe this has improved the quality and clarity of our work. If these updates and changes answer the reviewers (and other reviewers) points well, we would kindly ask if they would consider updating their score to reflect this.

---

### Official Review · Reviewer_WHwu · 2025-11-10

**Soundness:** 3
**Presentation:** 3
**Contribution:** 3
**Rating:** 6
**Confidence:** 4

**Summary:**

The paper introduces GABI (Geometric Autoencoder Priors for Bayesian Inversion), a framework for solving inverse problems, such as reconstructing temperature, pressure, or flow fields from limited noisy measurements, when system geometries vary. Instead of assuming known physics or PDEs, GABI learns a geometry-aware prior via an autoencoder that compresses physical fields across diverse geometries into a shared latent space. During inference, it performs Bayesian inversion in this latent space using Approximate Bayesian Computation (ABC) sampling, which scales efficiently on GPUs and outputs well-calibrated uncertainty. The approach enables a “train once, use anywhere” model that generalizes to unseen geometries and observation setups. Experiments across heat transfer, airfoil flow, car acoustics, and large-scale terrain flow show GABI matches supervised baselines in accuracy while offering full Bayesian uncertainty and far greater flexibility.

**Strengths:**

I think what makes this paper stand out isn’t just that it “does Bayesian inversion with deep learning”, but how it redefines the workflow. Most prior works either fix geometry and bake the PDE into the model, or treat inference as a supervised regression from observations to fields. Both approaches that crumble when geometry or observation setup changes. GABI breaks this dependency entirely: it learns a geometry-conditioned latent prior that can be reused across arbitrary domains, making it like an attempt for a foundation model for physical inversion. The theoretical guarantee that latent-space inference is equivalent to inference in the original field space is elegant and nontrivial, anchoring the method in solid Bayesian footing, rather than just heuristics.

Another secondary (yet distinctive) strength is the combination of MMD-based autoencoding and ABC sampling, which replaces the usual VAE–MCMC pairing with something GPU-parallelizable and architecture-agnostic, making large-scale problems (like the multi-GPU terrain flow case) actually tractable.

Finally, GABI’s decoupling of learning from the observation process is a rare and important contribution (it may have been proposed before but I am not 100% sure, other reviewers could correct me if I am wrong), most real engineering problems involve messy, variable sensors and noise models, and this paper’s “learn once, observe later” philosophy directly addresses that reality instead of hand-waving around it.

**Weaknesses:**

The paper seems to lean heavily on the assumption that a single latent space can meaningfully represent diverse geometries, but offers little analysis of when this assumption holds or how latent alignment behaves as geometry complexity scales, especially in high-frequency, discontinuous regimes (e.g., shocks or turbulence).

**Questions:**

How robust is the learned latent geometry field representation when the target system exhibits physical behaviors outside the training manifold, such as topological changes, discontinuous boundaries, or new forcing regimes? And can GABI’s prior adapt online without retraining to preserve its Bayesian consistency?

---

> ### Author Response · Authors · 2025-11-19
> **Response: latent space interpretation and out-of-distribution behaviour**
>
> We are very grateful for the reviewer's comments and feedback.
>
> i) **The paper seems to lean heavily on the assumption that a single latent space can meaningfully represent diverse geometries, but offers little analysis of when this assumption holds or how latent alignment behaves as geometry complexity scales, especially in high-frequency, discontinuous regimes (e.g., shocks or turbulence).**
>
> Indeed, we can only provide heuristics to assess whether a model performs well at constructing this latent space on a given dataset. We write these in Section 4:
>
> *In training the latent prior for GABI, the main way we have of knowing if the autoencoder networks perform adequately for the inference task, is by assessing the quality of reconstruction from a holdout set, as well as how close the embed holdout set is to being Gaussian in the latent space, which can easily be assessed qualitatively and quantitatively.*
>
> We would like to highlight that the car geometry and terrain geometry hold many discontinuities and sharp edges, but of course these are in-distribution for the test set.
>
> As for shocks, all indications are that GNNs can deal quite well will such discontinuities in the solution fields as these may in fact be highly predictable (eg. shocks around supersonic planes). As we do not solve the PDE or use PDE residuals in this framework we expect GABI would work with such problems. Turbulence on the other hand is very difficult to predict because of its chaotic sensitivity on geometry; even the slightest changes in geometry can induce extreme changes in solution fields. However, in such cases one is rarely interested in recovering the actual turbulence field, but usually some aggregate behaviour (either averaged in time or in space). These quantities behave much more nicely than fully developed turbulence. In fact in the airfoil example we use a Reynolds averaged Navier-Stokes dataset, Fig 10 for example, one can see the area of low pressure under the airfoil, and GABI's UQ increased uncertainty in these areas as the flow is known to be more unsteady there.
>
>
> ii) **How robust is the learned latent geometry field representation when the target system exhibits physical behaviors outside the training manifold, such as topological changes, discontinuous boundaries, or new forcing regimes? And can GABI’s prior adapt online without retraining to preserve its Bayesian consistency?**
>
> We hope our response to i) has answered the first part of this question. If not, please let us know how it would be useful for us to elaborate.
>
> We note the inlet conditions for the airfoil dataset and terrain dataset do change per example, however the training set is representative of the test set.
>
> In an effort to explore GABI's out-of-distribution (OOD) capabilities, we have added an additional airfoil experiment in section B.4.2. We take 10 unseen OOD airfoil shapes (OOD based on key airfoil statistics) and reconstruct flows based on 20 observed nodes. We see mild but limited robustness to outliers. We also note that eventually, the miss-specification of the prior for such out-of-distribution geometries may lead to unreliable inference.
>
>
> As for the second part, as it currently stands we do not have a way of adapting the GABI prior without retraining. Of course, one can have a “warm start” initializing a new model with the weights of the old model, which should in principle accelerate convergence.
>
> Please find attached the updated manuscript. We believe this has improved the quality and clarity of our work. If these updates and changes answer the reviewers (and other reviewers) points well, we would kindly ask if they would consider updating their score to reflect this.

---

### Author Response · Authors · 2025-11-19
**General Response to Reviewers**

We thank the reviewers for their time and care in reading our manuscript. In response to these comments and questions, we have made a number of changes to our manuscript. These include:

- Sec 1.1 (p.2): Added clarification and discussion around the observation operator $\mathbf{H}_o$.
- Sec 1.1 (p.3): Added overview description of the proposed methodology.
- Sec 2.1 (p.4): Added description and interpretation of Lemma 2.1.
- Sec 2.2 (p.5): Notational clarifications in  methodology.
- Sec 3 (p.6): Added link between classical Bayesian inversion and GP regression.
- Sec 4 (p.6): Added description of numerics, runtimes, and assessing candidate Autoencoders for inference.
- Sec 4.1 (p.7): Added discussion on interpretation of Figure 2.
- Appendix B.3.2: Added ablation study on the effect of network size on ABC and MCMC GABI inference.
- Appendix B.4.2: Added a study on the performance of GABI on out-of-distribution airfoil shapes.

We would like to kindly invite the reviewers to consider these changes, which we believe have improved the clarity of communication and interpretation of our work and results. We would be grateful if the reviewers would consider updating your evaluations if they feel these changes address their comments.

---

### Meta-Review · Area_Chair_uPYL · 2025-12-09

**Summary:**

The reviewers were, in average, positive about this submission. In some cases, their written comments are indeed more positive than captured by the numerical scores they gave. After reading the reviews, your responses, and the paper itself, I am following their recommendation. The paper provides an interesting approach to embedding priors over dynamical systems into an auto encoder, so that they can be efficiently `queried' for solving inverse problems for particular geometry. The methodology is novel and interesting, and could have numerous applications.

Effectively, one can think of this work as a conditional density estimator learned over many model-simulations, where one can then condition on the underlying geometry.  Conceptually, this seems related with a lot of recent work on learning amortised bayesian inference procedures from simulations, and it might have been useful to discuss these relationships more clearly (which is by no means to imply that this work is not novel-- indeed, and as pointed out by reviewers, it provides a highly original departure from usual approaches).

Please do take the input of the reviewers into account when preparing your final submission.

**Reviewer Concerns:**

Please see the meta-review about and their reviews. In particular:

There are some questions about generalisability to other domains, i.e. whether a latent representation is sufficient for a wide range of domains and problems. I share these concerns, and even if they can not be fully addressed in the paper, just a clear statement of limitations is important. I do think that the discussion of this (e.g. see WHwu and WWiD) is important, so this should be reflected in the final paper.

Reviewer TrCV found the paper hard to understand-- while I do not think their review is a ground for rejection, I do agree that it would benefit the authors (and leader to a paper which can have much more impact) if they would take their concerns seriously, and try to improve the presentation of the paper.  It would also be important to include some more details on computational cost.

**Reviewer Scores:**

I am unwilling to follow this request. The ACs are being asked to perform an unreasonable amount of extra work this year. In particular, I understand the concern about potential collusion. However, I do think that asking ACs to now read a whole new set of papers, reviews and (extremely long...) rebuttals and to try to condense them into decisions without a chance for discussions is both asking a lot from us, and will, inevitably, result in (on average) poorer decisions for everyone.

I do not see any value in spending even more time to try to 'guess' how each reviewer would have changed their mind or not. I am just trying to make appropriate decisions on the paper and focus on the science rather than wasting (additional) time on trying to emulate reviewers. The authors can see the reviewer inputs and should take it into account.

---

### Decision · Program_Chairs · 2026-01-26

Accept (Poster)